# Irreversibility in dynamical phases and transitions

Daniel S. Seara [1,2✉], Benjamin B. Machta[1,2✉] & Michael P. Murrell [1,2,3✉]

Living and non-living active matter consumes energy at the microscopic scale to drive emergent, macroscopic behavior including traveling waves and coherent oscillations. Recent work has characterized non-equilibrium systems by their total energy dissipation, but little has been said about how dissipation manifests in distinct spatiotemporal patterns. We introduce a measure of irreversibility we term the entropy production factor to quantify how time reversal symmetry is broken in field theories across scales. We use this scalar, dimensionless function to characterize a dynamical phase transition in simulations of the Brusselator, a prototypical biochemically motivated non-linear oscillator. We measure the total energetic cost of establishing synchronized biochemical oscillations while simultaneously quantifying the distribution of irreversibility across spatiotemporal frequencies.

---

[1] Department of Physics, Yale University, New Haven, CT 06511, USA. [2] Systems Biology Institute, Yale University, West Haven, CT 06516, USA. [3] Department of Biomedical Engineering, Yale University, New Haven, CT 06511, USA. ✉email: daniel.seara@yale.edu; benjamin.machta@yale.edu; michael.murrell@yale.edu

In many-body systems, collective behavior that breaks time-reversal symmetry can emerge due to the consumption of energy by the individual constituents[1–3]. In biological, engineered, and other naturally out of equilibrium processes, entropy must be produced so as to bias the system in a forward direction[4–9]. This microscopic breaking of time reversal symmetry can manifest at different length and time scales in different ways. For example, bulk order parameters in complex reactions can switch from exhibiting incoherent, disordered behavior to stable static patterns[10,11] or traveling waves of excitation[12,13] that break time reversal symmetry in both time and space simply by altering the strength of the microscopic driving force. Recent advances in stochastic thermodynamics have highlighted entropy production as a quantity to measure a system's distance from equilibrium[14–19]. While much work has been done investigating the critical behavior of entropy production at continuous and discontinuous phase transitions[20–28], dynamical phase transitions in spatially extended systems have only recently been investigated, and to date no non-analytic behavior in the entropy production has been observed[29,30].

To address this, we introduce what we term the entropy production factor (EPF), a dimensionless function of frequency and wavevector that measures time reversal symmetry breaking in a system's spatial and temporal dynamics. The EPF is a strictly non-negative quantity that is identically zero at equilibrium, quantifying how far individual modes are from equilibrium. Integrating the EPF produces a lower bound on the entropy production rate (EPR) of a system. We illustrate how to calculate the EPF directly from data using the analytically tractable example of Gaussian fields obeying partly relaxational dynamics supplemented with out of equilibrium coupling[31]. We then turn to the Brusselator reaction-diffusion model for spatiotemporal biochemical oscillations to study the connections between pattern formation and irreversibility. As the Brusselator undergoes a Hopf bifurcation far from equilibrium, its behavior transitions from incoherent and localized to coordinated and system-spanning oscillations in a discontinuous transition. The EPF quantifies the shift in irreversibility from high to low wavenumber as this transition occurs, but the EPR is indistinguishable from that of the well-mixed Brusselator where synchronization cannot occur. Importantly, the EPF can be calculated in any number of spatial dimensions, making it broadly applicable to a wide variety of data types, from particle tracking to 3+1 dimensional microscopy time series.

## Results

**Entropy production factor derivation.** Consider a system described by a set of $M$ real, random variables obeying some possibly unknown dynamics. A specific trajectory of the system over a total time $T$ is given by $\mathbf{X} = \{X^i(t) | t \in [0, T]\}$. Given an ensemble of stationary trajectories, the average EPR, $\dot{S}$, is bounded by[5,6,32]

$$\dot{S} \geq \lim_{T \to \infty} \frac{1}{T} D_{\mathrm{KL}}\left(P[\mathbf{X}] \parallel P[\widetilde{\mathbf{X}}]\right);$$

$$D_{\mathrm{KL}}\left(P[\mathbf{X}] \parallel P[\widetilde{\mathbf{X}}]\right) = \left\langle \log\left(\frac{P[\mathbf{X}]}{P[\widetilde{\mathbf{X}}]}\right) \right\rangle_{P[\mathbf{X}]} \quad (1)$$

where we have set $k_{\mathrm{B}} = 1$ throughout and $D_{\mathrm{KL}}$ denotes the Kullback–Leibler divergence which measures the distinguishability between two probability distributions. $P[\mathbf{X}]$ and $P[\widetilde{\mathbf{X}}]$ are the steady-state probability distribution functionals of observing the path $\mathbf{X}(t)$ of length $T$ and the probability of observing its reverse path, respectively. Therefore, the KL divergence in Eq. (1) measures the statistical irreversibility of a signal, and saturates the

bound when $\mathbf{X}$ contains all relevant, non-equilibrium degrees of freedom.

We further bound the irreversibility itself by assuming the paths obey a Gaussian distribution. Writing the Fourier transform of $X^i(t)$ as $x^i(\omega)$, where $\omega$ is the temporal frequency, and writing the column vector $\mathbf{x}(\omega) = (x^1(\omega), x^2(\omega), \dots)^T$:

$$P[\mathbf{x}(\omega)] = \frac{1}{Z} \prod_{\omega_n} \exp\left(-\frac{1}{2T} \mathbf{x}^\dagger C^{-1} \mathbf{x}\right), \quad (2)$$

where $\mathbf{x}^\dagger$ denotes the conjugate transpose of the vector $\mathbf{x}$ evaluated at the discrete frequencies $\omega_n = 2\pi n T^{-1}$. $C(\omega_n)$ is the covariance matrix in Fourier space with elements $C^{ij}(\omega_n) = \langle x^i(\omega_n) x^j(-\omega_n) \rangle T^{-1}$, and $Z$ is the partition function. The expression for $P[\widetilde{x}]$ is identical but with $C^{-1}(\omega_n) \to C^{-1}(-\omega_n)$ [see Supplementary Note 1]. Combining Eq. (1) with Eq. (2) and taking $T \to \infty$, we arrive at our main result:

$$\dot{S} = \int \frac{d\omega}{2\pi} \, \mathcal{E}(\omega); \qquad \mathcal{E}(\omega) = \frac{1}{2} \left[ C^{-1}(-\omega) - C^{-1}(\omega) \right]_{ij} C^{ji}(\omega). \quad (3)$$

This defines the EPF, $\mathcal{E}(\omega)$, which measures time reversal symmetry breaking interactions between $M \geq 2$ variables, while integrating $\mathcal{E}$ gives $\dot{S}$. $\mathcal{E}(\omega) = D_{\mathrm{KL}}(P[\mathbf{x}(\omega)] \parallel P[\widetilde{\mathbf{x}}(\omega)])$ measures the Kullback–Leibler divergence between the joint distribution of $M$ modes at a single frequency $\omega$. While this quantity does not scale with trajectory length, the density of modes near a particular frequency is related to the total trajectory time by $\Delta\omega = 2\pi T^{-1}$. Since $\pm \omega$ modes must be complex conjugates of each other and an overall average phase is prohibited by time translation invariance, asymmetry between these distributions can only be captured by relative phase relationships, quantified by their correlation functions. $\mathcal{E}$ is large when one variable tends to lead another in phase, implying a directed rotation between these variables in the time domain.

As mentioned above, $P[\mathbf{x}(\omega)]$ describes the dynamics of a non-equilibrium steady state, and no reversal of external protocol is assumed. Further, in writing an expression for $P[\widetilde{\mathbf{x}}(\omega)]$, we assume that the observables are scalar, time-reversal symmetric quantities, such as the chemical concentrations we analyze below.

The Gaussian assumption we make here makes Eq. (3) exact only for systems obeying linear dynamics. Nevertheless, $\mathcal{E}$ is still defined for non-linear systems, where the integrated $\mathcal{E}$ lower bounds the true $\dot{S}$. To see this, consider projecting complex dynamics onto Gaussian dynamics by choosing a data processing procedure which preserves two point correlations but which removes higher ones. This can be accomplished by multiplying every frequency by an independent random phase — a post-processing procedure which can be applied to individual trajectories. Post-convolution, the integrated EPF is equal to the KL divergence rate between forward and backwards rates. From the data processing inequality, the KL divergence rate of the true fields must be higher, so that the integrated EPF lower bounds the true entropy production rate [see Supplementary Note 2]. In addition to bounding the true $\dot{S}$, we expect the integral of $\mathcal{E}$ to be a good approximation for the wide class of systems where linearization is reasonable. Such Gaussian approximations are starting points in many field theories, with higher order interactions accounted for by adding anharmonic terms in the action of Eq. (2). While this is not our focus here, we expect these additional terms to systematically capture corrections to $\dot{S}$ that do not appear in Eq. (3). As $C^{ii}(\omega) = C^{ii}(-\omega)$, the only contributions to $\mathcal{E}$ come from the cross-covariances between the random variables of interest. As such, this bound yields exactly 0 for a

single variable even though higher order terms may contribute to $\dot{S}$.

This formulation extends naturally to random fields. For $M$ random fields in $d$ spatial dimensions, $\boldsymbol{\phi} = \{\phi^i(\mathbf{x},t)|t \in [0,T], \mathbf{x} \in \mathbb{R}^d\}$, the EPR density, $\dot{s} \equiv \dot{S}/V$ where $V$ is the system volume, is:

$$\dot{s} = \int \frac{d\omega}{2\pi} \frac{d^d\mathbf{q}}{(2\pi)^d} \mathcal{E}(\mathbf{q},\omega);$$
$$\mathcal{E}(\mathbf{q},\omega) = \frac{1}{2}\left[C^{-1}(\mathbf{q},-\omega) - C^{-1}(\mathbf{q},\omega)\right]_{ij} C^{ji}(\mathbf{q},\omega). \quad (4)$$

where $C^{ij}(\mathbf{q},\omega)$ is the dynamic structure factor and $\mathcal{E}(\mathbf{q},\omega)$ is now a function of wavevector $q$ and frequency $\omega$ [see Supplementary Note 1].

Even without an explicit, analytic expression for the structure factor, $C$, we can estimate $\mathcal{E}$ from data. To use Eq. (4), we consider data of $N$ finite length trajectories of $M$ variables over a time $T$ in $d$ spatial dimensions. Each dimension has a length $L_i$. We create an estimate of the covariance matrix, $\widetilde{C}(\mathbf{q},\omega)$, from time-series using standard methods [see Methods]. These measurements will inevitably contain noise that is not necessarily time-reversal symmetric, even for an equilibrium system. Noise due to thermal fluctuations and finite trajectory lengths in the estimate of $\widetilde{C}$ from a single experiment ($N=1$) will systematically bias our estimated $\mathcal{E}$ by $\Delta\mathcal{E} = \frac{M(M-1)}{2}$ at each frequency and will thereby introduce bias and variance in our measurement of $\dot{s}$. We can simply remove the bias from our measured $\mathcal{E}$, but to reduce the variance, we smooth $\widetilde{C}$ by component-wise convolution with a multivariate Gaussian of width $\boldsymbol{\sigma} = (\sigma_{q_1}, \ldots, \sigma_{q_d}, \sigma_\omega)$ in frequency space, giving $\hat{C}$. This is equivalent to multiplying each component of the time domain $\widetilde{C}(\mathbf{r},t)$ by a Gaussian, cutting off the noisy tails in the real space covariance functions at large lag times. We then use $\hat{C}$ in Eq. (4) to create our final estimator for the EPF, $\hat{\mathcal{E}}$, and thereby the EPR, $\hat{s}$. We calculate and remove the bias in $\hat{\mathcal{E}}$ and $\hat{s}$ in all results below [see Methods]. Smoothing $\widetilde{C}$ with increasingly wide Gaussians in $\omega$ and $\mathbf{q}$ leads to a systematic decrease in $\hat{s}$ due to reduced amplitudes in $\widetilde{C}$ (Supplementary Notes 3 and 4, Supplementary Fig. 1).

To illustrate the information contained in $\mathcal{E}$, its numerical estimation, and the accuracy of $\hat{s}$, we analyze simulations of coupled, one-dimensional Gaussian stochastic fields for which $\mathcal{E}$ and $\dot{s}$ can be calculated analytically. We then study simulations of the reaction-diffusion Brusselator, a prototypical model for non-linear biochemical oscillators, and use $\mathcal{E}$ to study how irreversibility manifests at different time and length scales as the system undergoes a Hopf bifurcation[33].

**Driven Gaussian fields.** Consider two fields obeying Model A dynamics[31] with non-equilibrium driving parametrized by $\alpha$:

$$\partial_t\phi(x,t) = -D\frac{\delta\mathcal{F}}{\delta\phi} - \alpha\psi + \sqrt{2D}\xi_\psi$$
$$\partial_t\psi(x,t) = -D\frac{\delta\mathcal{F}}{\delta\psi} + \alpha\phi + \sqrt{2D}\xi_\phi, \quad (5)$$

where $\xi(\mathbf{x},t)$ is Gaussian white noise with variance $\langle \xi^i(\mathbf{x},t)\xi^j(\mathbf{x}',t')\rangle = \delta^{ij}\delta(\mathbf{x}-\mathbf{x}')\delta(t-t')$, $D$ is a relaxation constant, and $\delta\mathcal{F}/\delta\phi$ is the functional derivative with respect to $\phi$ of the free energy $\mathcal{F}$ given by:

$$\mathcal{F} = \int dx \left[\frac{r}{2}(\phi^2 + \psi^2) + \frac{1}{2}\left(|\partial_x\phi|^2 + |\partial_x\psi|^2\right)\right], \quad (6)$$

so that the fields have units of $\ell^{1/2}$ and $r$ penalizes large amplitudes.

The EPR density, $\dot{s}$, is calculated analytically in two ways. First, we solve Eq. (1) directly using the Onsager–Machlup functional for the path probability functional of $\boldsymbol{\eta}(x,t) = (\phi(x,t), \psi(x,t))^T$[4,34]. Second, the covariance matrices are calculated analytically, used to find $\mathcal{E}$ through Eq. (4), and integrated to find $\dot{s}$. Both cases give the same result for $\dot{s}$. The result for both $\mathcal{E}$ and $\dot{s}$ are [see Supplementary Note 5]:

$$\mathcal{E}^{\mathrm{DGF}} = \frac{8\alpha^2\omega^2}{(\omega^2 - \omega_0^2(q))^2 + (2D(r+q^2)\omega)^2}, \quad \dot{s}^{\mathrm{DGF}} = \frac{\alpha^2}{D\sqrt{r}}. \quad (7)$$

We see that $\mathcal{E}^{\mathrm{DGF}} \geq 0$ and exhibits a peak at $(q,\omega) = (0,\omega_0(0))$, where $\omega_0(q) = \sqrt{(D(r+q^2))^2 + \alpha^2}$, indicating that the system is driven at all length scales with a driving frequency of $\alpha$, dampened by an effective spring constant $Dr$. In addition, it is clear that multiple combinations of $\alpha$, $r$, and $D$ can give the same value for $\dot{s}$ while $\mathcal{E}$ distinguishes between equally dissipative trajectories in the shape and location of its peaks. In this way, $\mathcal{E}$ gives information about the form of the underlying dynamics not present in the total EPR. We note that $\mathcal{E}^{\mathrm{DGF}}$ is also recovered using an appropriately modified version of the generalized Harada–Sasa Relation introduced in 34 [see Supplementary Note 6].

We perform simulations to assess how well $\mathcal{E}$ can be extracted from time series data of fields [see methods for details]. The estimated $\hat{\mathcal{E}}$ shows excellent agreement with Eq. (7) (Fig. 1). Integrating $\hat{\mathcal{E}}$ gives $\hat{s}$, which also shows good agreement with $\dot{s}^{\mathrm{DGF}}$.

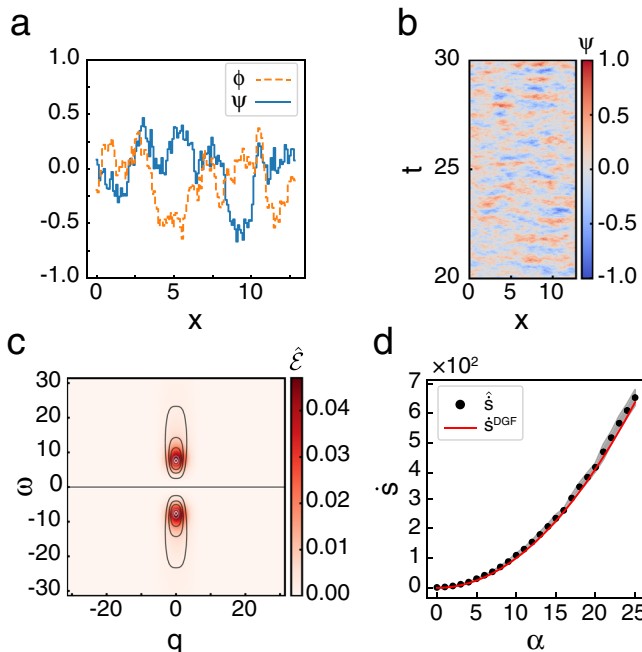

**Fig. 1 Entropy production rate and entropy production factor are well estimated for driven Gaussian fields. a** Snapshot of typical configurations of both fields, $\psi$ (blue solid line) and $\phi$ (orange dashed line) obeying Eq. (5) for $\alpha = 7.5$. **b** Subsection of a typical trajectory for one field for $\alpha = 7.5$ in dimensionless units. Colors indicate the value of the field at each point in spacetime. **c** $\hat{\mathcal{E}}$ for $\alpha = 7.5$ averaged over $N = 10$ simulations. Contours show level sets of $\mathcal{E}^{\mathrm{DGF}}$. **d** Measured $\dot{s}$ vs. $\alpha$ for simulations of total time $T = 50$ and length $L = 12.8$. Red line shows the theoretical value, $\dot{s}^{\mathrm{DGF}}$. Mean ± s.d. of $\hat{s}$ given by black dots and shaded area. See Supplementary Table 1 for all simulation parameters.

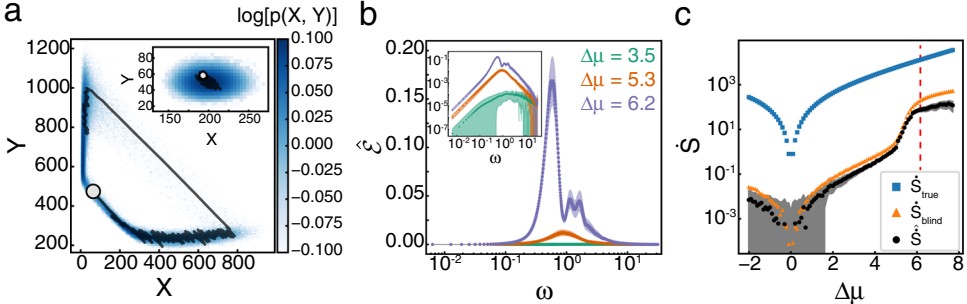

**Fig. 2 $\dot{S}$ and $\mathcal{E}$ for well-mixed Brusselator. a** Typical trajectory in $(X, Y)$ space for $\Delta\mu = 6.2$. The occupation probability distribution is shown in blue, with a subsection of a typical trajectory shown in black. The end of the trajectory is marked by the white circle. Inset shows the same information for the system at equilibrium, where $\Delta\mu = 0$, with the same colorbar as the main figure. **b** $\hat{\mathcal{E}}$ for $\Delta\mu = [3.5, 5.3, 6.2]$ shown in green, orange, and purple, respectively. Shaded area shows mean ± s.d. of $\hat{\mathcal{E}}$ for $N = 50$ simulations. $\hat{\mathcal{E}}$ is symmetric in $\omega$, so only the positive axis is shown. Inset shows the same curves on a log-log scale. **c** $\dot{S}$ as a function of $\Delta\mu$. Blue squares, orange triangles, and black circles show results for $\dot{S}_{\text{true}}$, $\dot{S}_{\text{blind}}$, and $\hat{\dot{S}}$, respectively. Shaded area shows mean ± s.d. of $\hat{\dot{S}}$ for $N = 50$ simulations. Vertical red dashed line indicates $\Delta\mu_{\text{HB}}$. See Supplementary Table 2 for all simulation parameters.

Our estimator gives exact results for the driven Gaussian fields because the true path probability functional for these fields is Gaussian. In contrast, the complex patterns seen in nature arise from systems obeying highly non-linear dynamics. For such dynamics, our Gaussian approximation is no longer exact but provides a lower bound on the total irreversibility. To investigate how irreversibility correlates with pattern formation, we study simulations of the Brusselator model for biochemical oscillations[35]. We begin by describing the various dynamical phases of the equations of motion. Next, we calculate $\mathcal{E}$ and $\dot{S}$ for only the reactions before adding diffusion to study the synchronized oscillations that arise in the one-dimensional reaction-diffusion system.

**Reaction-diffusion Brusselator.** We use a reversible Brusselator model[30,35–37] with dynamics governed by the reaction equations:

$$A \underset{k_1^-}{\overset{k_1^+}{\rightleftharpoons}} X; \quad B + X \underset{k_2^-}{\overset{k_2^+}{\rightleftharpoons}} Y + C; \quad 2X + Y \underset{k_3^-}{\overset{k_3^+}{\rightleftharpoons}} 3X; \quad (8)$$

where $\{A, B, C\}$ are external chemical baths with fixed concentrations $\{a, b, c\}$, and all the reactions occur in a volume $V$ (Fig. 2a). The system is in equilibrium when the external chemical baths and reaction rates obey $bk_2^+ k_3^+ = ck_2^- k_3^-$. When this equality is violated, the system is driven away from equilibrium and exhibits cycles in the $(X, Y)$ plane. Defining

$$\Delta\mu = \log\left(\frac{bk_2^+ k_3^+}{ck_2^- k_3^-}\right), \quad (9)$$

the Brusselator is at equilibrium when $\Delta\mu = 0$ and is driven into a non-equilibrium steady state when $\Delta\mu \neq 0$. We vary $b$ and $c$ to change $\Delta\mu$ while keeping the product $(bk_2^+ k_3^+)(ck_2^- k_3^-) = 1$, keeping the rate at which reactions occur constant for all $\Delta\mu$[38].

As $\Delta\mu$ increases, the macroscopic version of Eq. (8) undergoes dynamical phase transitions. For all $\Delta\mu$, there exists a steady state $(X_{\text{ss}}, Y_{\text{ss}})$, the stability of which is determined by the relaxation matrix, $R$ [Supplementary Note 7]. The two eigenvalues of $R$, $\lambda_\pm$, divide the steady state into four classes[33]:

- $\lambda_\pm \in \mathbb{R}_{<0} \rightarrow$ Stable attractor, no oscillations
- $\lambda_\pm \in \mathbb{C}$, $\text{Re}[\lambda_\pm] < 0 \rightarrow$ Stable focus
- $\lambda_\pm \in \mathbb{C}$, $\text{Re}[\lambda_\pm] > 0 \rightarrow$ Hopf Bifurcation, limit cycle
- $\lambda_\pm \in \mathbb{R}_{>0} \rightarrow$ Unstable repeller

The eigenvalues undergo these changes as $\Delta\mu$ changes, allowing us to consider $\Delta\mu$ as a bifurcation parameter. We define $\Delta\mu_{\text{HB}}$ as the value of $\Delta\mu$ where the macroscopic system undergoes the Hopf bifurcation.

Non-equilibrium steady states are traditionally characterized by their circulation in a phase space[39–43]. One may then question how it is possible to detect non-equilibrium effects in the Brusselator when the system's steady state is a stable attractor with no oscillatory component. While this is true for the macroscopic dynamics used to derive $\lambda_\pm$, we simulate a system with finite numbers of molecules subject to fluctuations. These stochastic fluctuations give rise to circulating dynamics, even when the deterministic dynamics do not[36]. We see persistent circulation in the $(X, Y)$ plane when $\lambda_\pm \in \mathbb{R}_{<0}$, with the vorticity changing sign around $\Delta\mu = 0$ (Supplementary Fig. 2).

In order to assess the accuracy of our estimated EPR, $\hat{\dot{S}}$, we calculate an estimate of the true EPR, $\dot{S}_{\text{true}}$, for a simulation of Eq. (8) by calculating the exact entropy produced by each reaction that occurs in the trajectory[44], and then fitting a line to the cumulative sum (Supplementary Fig. 3, Methods). We find that $\hat{\dot{S}}$ significantly underestimates $\dot{S}_{\text{true}}$ (note the logged axes in Fig. 2c) due to the Brusselator's hidden dynamics. In the Brusselator, information is lost because the observed trajectories are coarse-grained — they do not distinguish between reactions that take place forward through the second reaction or backwards through the third reaction in Eq. (8). These pathways would be distinguishable if trajectory of $B$ and $C$ were also observable.

Our method relies purely on system dynamics to give $\hat{\dot{S}}$. Eq. (1) is true only if all microscopic details are captured by trajectories $\mathbf{X}$. If $\mathbf{X}$ is already coarse-grained, multiple microscopic trajectories will be indistinguishable and Eq. (1) will underestimate the true entropy production rate due to the data processing inequality[19,45,46].

In order to account for this, we recalculate $S$ by considering the rate at which a given transition can occur as the sum over all chemical reactions that give the same dynamics [see Methods]. For example, a transition from $(X, Y) \rightarrow (X - 1, Y + 1)$ can occur via reaction $k_2^+$ or $k_3^-$ in the Brusselator, each of which produces a different amount of entropy in general. Looking only in the $(X, Y)$ plane, it is impossible to tell which reaction took place. When calculating the entropy produced by only the observable dynamics, the rate of making the transition $(X, Y) \rightarrow (X - 1, Y + 1)$ is $k_f = k_2^+ + k_3^-$, while the rate of making the reverse transition is $k_r = k_2^- + k_3^+$, and the entropy produced is $\log\frac{k_f}{k_r}$. This estimate of the EPR, which we name $\dot{S}_{\text{blind}}$, is a coarse-graining of $\dot{S}_{\text{true}}$, giving the relation $\dot{S}_{\text{blind}} \leq \dot{S}_{\text{true}}$[47]. We find that $\dot{S}_{\text{blind}}$ shows excellent agreement with $\hat{\dot{S}}$, indicating that the Gaussian approximation provides a good estimate for the observable dynamics even when the system is highly non-linear.

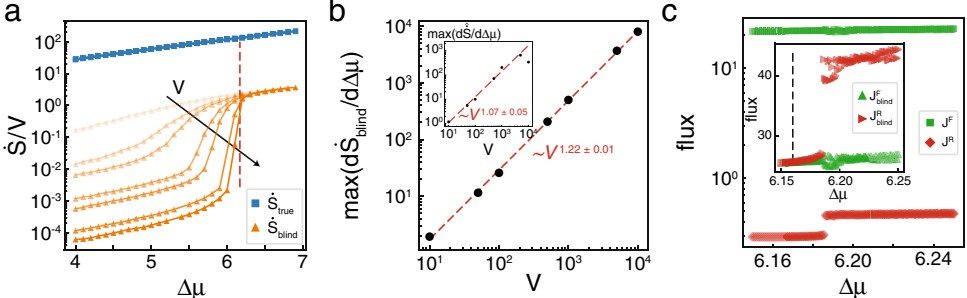

**Fig. 3 Finite size scaling of $\dot{S}$. a** $\dot{S}_{true}/V$ (blue squares) and $\dot{S}_{blind}/V$ (orange triangles) for system volumes $V$ = [10, 50, 100, 500, 1000, 5000, 10000], showing an increasingly sharp transition in $\dot{S}_{blind}$, but not in $\dot{S}_{true}$. $\dot{S}_{blind}$ shows no volume dependence below the transition, and is linear dependent on $V$ above it. Vertical red dashed line shows $\Delta\mu_{HB}$. **b** Maximum value of $\partial\dot{S}_{blind}/\partial\Delta\mu$ shows a power-law dependence with volume. Inset shows the same measurement for $\partial\hat{S}/\partial\Delta\mu$. **c** Forward and reverse fluxes, $J^F$ (green squares) and $J^R$ (red diamonds), obtained from numerical integration of deterministic equations of motion for the Brusselator. Inset shows $J^F_{blind}$ (green upright triangles) and $J^R_{blind}$ (red rightward triangles). Vertical black dashed line shows $\Delta\mu_{HB}$.

To further benchmark our estimator, we calculate $\dot{S}$ using two alternative methods, one based on the thermodynamic uncertainty relation (TUR)[7,48] and one based on measuring first passage times (MFPT)[49]. The prior method measures a macroscopic current based on a weighted average of a system's trajectory, $j_{\mathbf{d}}$, and estimates the EPR using the TUR for diffusive dynamics, $\dot{S} \geq 2\langle j_{\mathbf{d}}^2\rangle(\tau_{obs}\text{Var}[j_{\mathbf{d}}])^{-1}$, where $\langle\rangle$ and Var[] denote an ensemble average and variance taken after an observation time $\tau_{obs}$[17]. The latter method requires measuring the MFPT of an observable $\mathcal{O}$ constructed from the system's dynamics to reach a threshold that depends on a user-defined error tolerance. We choose $\mathcal{O}$ and the threshold based on a drift-diffusion approximation for the winding number of the Brusselator (Supplementary Note 8). Similarly to $\hat{S}$, both of these methods saturate to the true $\dot{S}$ for systems obeying linear dynamics. As such, they also approximate $\dot{S}_{blind}$, but we find that they provide a looser bound than $\hat{S}$ (Supplementary Fig. 4).

Prior to $\Delta\mu_{HB}$, both $\dot{S}$ and $\dot{S}_{blind}$ show a shift in their trends, but $\dot{S}_{true}$ does not. The smooth transition is due to the finite system size we employ, and gets sharper as a power law as the system gets larger (Fig. 3a). The power law exponent measured from $\hat{S}$ is nearly linear, consistent with the Gaussian assumption. The exponent differs from that of $\dot{S}_{blind}$ because our Gaussian assumption breaks down at the high values of $\Delta\mu$ where the maximum slope occurs (Fig. 3b).

The Hopf bifurcation for the Brusselator is supercritical[23], meaning the limit cycle grows continuously from the fixed point when $\Delta\mu - \Delta\mu_{HB} \ll 1$. Further from the transition point, the trajectory makes a discontinuous transition. At our resolution in $\Delta\mu$, this discontinuous transition is what underlies the shift in $\dot{S}_{blind}$ of the Brusselator. This same transition is present in $\dot{S}_{true}$, but is difficult to detect numerically for reasons we explain here. In the deterministic limit, $\dot{S}_{true} = \Delta\mu(J^F - J^R)$, where $J^F = b\langle x\rangle k_2^+$ and $J^R = c\langle y\rangle k_2^-$ are the forward and reverse fluxes for transforming a $B$ molecule into a $C$ molecule. $\langle x\rangle$ is a constant, but by numerically integrating the deterministic version for Eq. (8), we observe a discontinuity in $\langle y\rangle$ above the Hopf bifurcation. However, $J^F \gg J^R$, obscuring the discontinuity in $\dot{S}_{true}$ (Fig. 3c). Upon coarse-graining, we have $\dot{S}_{blind} = \Delta\mu(J^R_{blind} - J^F_{blind})$, with $J^F_{blind} = b\langle x\rangle k_2^+ + \langle x\rangle^3 k_3^-$ and $J^R = c\langle y\rangle k_2^- + \langle x\rangle^2\langle y\rangle k_3^+$. These two terms are equal to each other for $\Delta\mu < \Delta\mu_{HB}$ and diverge continuously when $\Delta\mu \gtrsim \Delta\mu_{HB}$, followed by the relatively large discontinuity in $J^R_{blind}$ (Fig. 3c, inset).

One gains further insight into the dynamics through the transition by studying $\hat{\mathcal{E}}$ (Fig. 2b). For $\Delta\mu < \Delta\mu_{HB}$, $\hat{\mathcal{E}}$ exhibits a

single peak that increases in amplitude while decreasing in frequency as $\Delta\mu$ increases. Above $\Delta\mu_{HB}$, the peak frequency makes a discontinuous jump, the magnitude of the peak grows rapidly, and additional peaks at integer multiples of the peak frequency appear due to the non-linear shape of the limit cycle attractor. These harmonics are expected for dynamics on a non-circular path. For $\Delta\mu < \Delta\mu_{HB}$, the magnitude of the peak is independent of system volume, while it gains a linear volume dependence in the limit cycle. The width of the peak is also maximized near the transition, reflecting a superposition of frequencies present in the trajectories (Supplementary Fig. 5).

To investigate how dynamical phase transitions manifest in the irreversibility of spatially extended systems, we simulate a reaction-diffusion Brusselator on a one-dimensional periodic lattice with $L$ compartments, each with volume $V$, spaced a distance $h$ apart. The full set of reactions are now

$$A_i \underset{k_1^-}{\overset{k_1^+}{\rightleftharpoons}} X_i; \quad B_i + X_i \underset{k_2^-}{\overset{k_2^+}{\rightleftharpoons}} Y_i + C_i; \quad 2X_i + Y_i \underset{k_3^-}{\overset{k_3^+}{\rightleftharpoons}} 3X_i;$$

$$X_i \underset{d_X}{\overset{d_X}{\rightleftharpoons}} X_{i+1}; \qquad Y_i \underset{d_Y}{\overset{d_Y}{\rightleftharpoons}} Y_{i+1}; \qquad i \in [1, L] \tag{10}$$

where $d_j = D_j/h^2$, and $D_j$ is the diffusion constant of chemical species $j = \{X, Y\}$. Qualitatively different dynamics occur based on the ratio $D_X/D_Y$. $D_X/D_Y \ll 1$ yields static Turing patterns[10,29]. We focus on the $D_X/D_Y \gg 1$ regime which exhibits dynamic, excitable waves. All values of $\{a_i, b_i, c_i\}$ are kept constant in each compartment.

In the steady state, the reaction-diffusion Brusselator has the same dynamics as the well mixed Brusselator, and so it is not surprising that it's EPR curve as a function of $\Delta\mu$ is similar (Supplementary Fig. 6). However, unlike the well-mixed system, the Hopf bifurcation signals the onset of qualitatively distinct dynamics in the reaction-diffusion system. Prior to the Hopf bifurcation, there are no coherent, spatial patterns in the system's dynamics (Fig. 4a). Above the Hopf bifurcation, system-spanning waves begin to emerge that synchronize the oscillations across the system (Fig. 4b). Following standard methods[50,51], we define the synchronization order parameter, $0 \leq r < 1$, using

$$re^{i\psi} = \frac{1}{T}\int_0^T dt \frac{1}{M}\sum_{j=1}^{M} e^{i\theta_j(t)} \tag{11}$$

where $\theta_j(t)$ is the phase of the oscillator at position $x_j$ and time $t$, $M$ is the number of oscillators (here, the number of lattice sites in our simulation), and $T$ is the temporal extent of the data [Methods]. $\psi$ denotes the overall phase, and $r$ is close to zero in the asynchronous phase and approaches one as the oscillators synchronize.

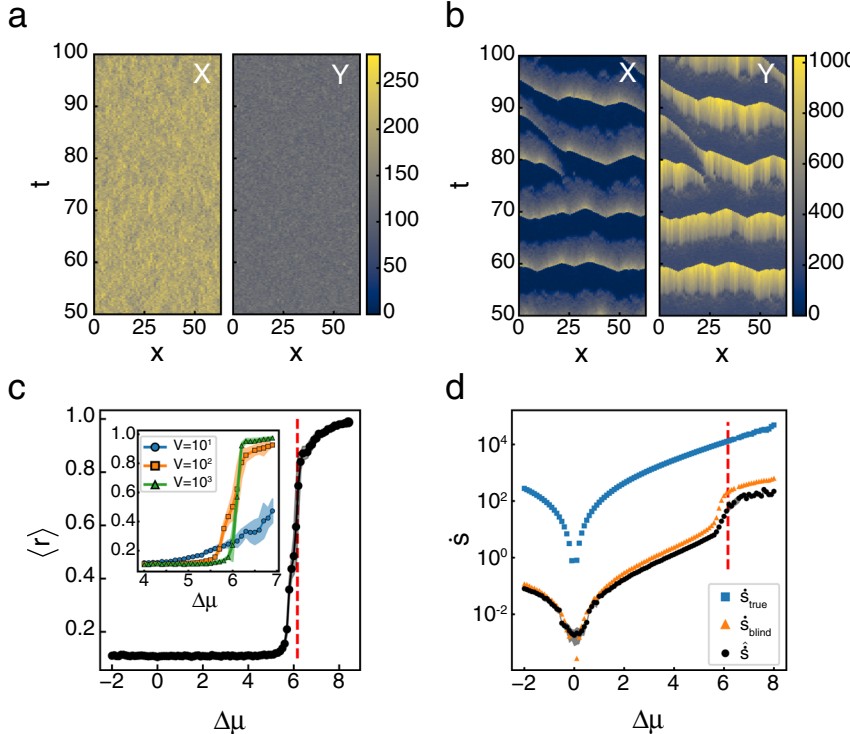

**Fig. 4 Reaction-diffusion Brusselator synchronizes above Hopf bifurcation. a** Subsection of a typical trajectory for $X(x, t)$ and $Y(x, t)$ for **a** $\Delta\mu = 3.5$, below the Hopf Bifurcation and **b** $\Delta\mu = 6.2$, above it. Color indicates the local number of the chemical species. **c** Synchronization order parameter, $\langle r \rangle$, as a function of $\Delta\mu$. Vertical red dashed line indicates $\Delta\mu_{HB}$. Inset shows the same measurement for volumes $V = \{10^1, 10^2, 10^3\}$ shown by blue circles, orange squares, and green triangles, respectively, at each lattice site over a smaller region of $\Delta\mu$. Dots and shaded areas show mean ± s.d. of $N = 10$ simulations. **d** $\dot{s}$ as a function of $\Delta\mu$. Blue squares, orange triangles, and black circles show results for $\dot{s}_{true}$, $\dot{s}_{blind}$, and $\dot{s}$, respectively. Shaded area shows mean ± s.d. of $N = 10$ simulations. Vertical dashed red line indicates $\Delta\mu_{HB}$. See Supplementary Tables 3 and 4 for all simulation parameters.

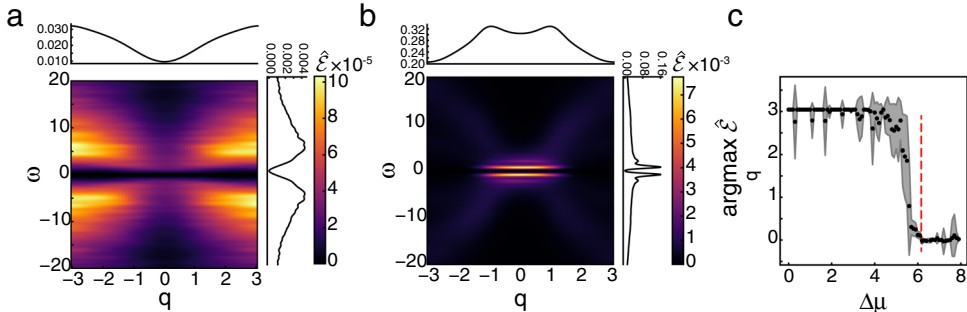

**Fig. 5 Entropy production factor and macroscopic dynamics. a** $\mathcal{E}$ averaged over $N = 10$ simulations for $\Delta\mu = 4.0$, i.e., $\Delta\mu < \Delta\mu_{HB}$. Line plots on top and left of figure show marginals over $\omega$ and $q$, respectively. **b** Similar to **a**, but for $\Delta\mu = 6.2$, i.e., $\Delta\mu > \Delta\mu_{HB}$. **c** Wavenumber, $q$, that maximizes $\hat{\mathcal{E}}$ as a function of $\Delta\mu$. Vertical dashed red line shows $\Delta\mu_{HB}$. Black dots and shaded area show mean ± s.d. over $N = 10$ simulations.

Below $\Delta\mu_{HB}$, $r$ is low and rapidly approaches one as the system approaches the macroscopic bifurcation point (Fig. 4c). Like $\dot{S}$, this transition occurs more sharply and closer to $\Delta\mu_{HB}$ as the system size increases, approaching the discontinuous transition to the limit cycle behavior (Fig. 4c, inset)[52]. Throughout these changes, the system is driven further from equilibrium, as reflected in the increasing $\hat{s}$ (Fig. 4d). The shift to collective behavior is not reflected in $\dot{s}$ as it is almost identical to $\dot{S}$ found for the well-mixed Brusselator (Supplementary Fig. 6). Instead, $\mathcal{E}$ carries the signature of the dynamical phase transition. For $\Delta\mu < \Delta\mu_{HB}$, $\hat{\mathcal{E}}$ shows peaks at high wavenumbers, reflecting that irreversibility is occurring incoherently over short length scales. Above $\Delta\mu_{HB}$, as the system shows synchronized oscillations, there is an abrupt shift in the peaks of $\hat{\mathcal{E}}$ to low $q$, indicating that this collective behavior carries the majority of the irreversibility

(Fig. 5b,c). We also infer that the collective behavior is partially composed of traveling waves due to the streaks in $\hat{\mathcal{E}}$ (Fig. 5b). The slight offset in the transition occurs for high values of $\Delta\mu < \Delta\mu_{HB}$ where small regions synchronize for short periods of time, but system wide oscillations are not observed (Supplementary Fig. 7). Furthermore, the transition moves closer to the macroscopic transition point with increased volume of the individual compartments (Supplementary Fig. 7).

## Discussion

Previous work has investigated the behavior of $\dot{S}$ at thermodynamic phase transitions with the work of [22] finding general signatures of discontinuous phase transitions in $\dot{S}$ which agree with our results. While [26] found $\dot{S}$ to have a discontinuity of its first derivative with respect to $\Delta\mu$ in a slightly modified version of

the well-mixed Brusselator, work on the same system presented here did not find any non-analytic behavior in $\dot{S}_{\text{true}}$[30]. We show that a discontinuous phase transition exists in our model, but the magnitude of the discontinuity is small and difficult to detect in $\dot{S}_{\text{true}}$ and is more easily seen in the coarse-grained $\dot{S}_{\text{blind}}$ (Fig. 3). Further, other spectral decompositions of the dissipation rate either assume a particular form for the underlying dynamics[27] or require the measurement of a response function in addition to the correlation function[34], which is often difficult to perform in experiments.

Here, we illustrated that the total irreversibility rate cannot distinguish between the dynamical phase transitions in the well-mixed and the spatially extended Brusselator (Supplementary Fig. 6). While the EPR quantifies the emergence of oscillations, the synchronization of the oscillations across space is only captured in $\mathcal{E}$ by its peak shifting from high to low wavenumber (Fig. 5). By simulating systems with increasing compartment volumes, this shift occurs closer to the macroscopic transition point (Supplementary Fig. 7), similarly to the increasing sharpness of the shift in $\dot{S}$ for the well-mixed Brusselator (Fig. 3). Thus, synchronization is intimately related to the emergence of oscillations. We hypothesize that synchronization occurs due the presence of a slow segment of the Brusselator dynamics (Fig. 2a). The time spent in the slow portion of the dynamics allows neighboring oscillators to reduce their relative phase through their diffusive coupling, allowing previously out-of-sync lattice sites to synchronize via the low-cost mechanism of diffusion. This is further seen by the higher value of $\dot{s}_{\text{blind}}$ for the reaction-diffusion Brusselator compared to $\dot{S}_{\text{blind}}$ for the well-mixed Brusselator when $\Delta\mu < \Delta\mu_{\text{HB}}$, but not for $\Delta\mu > \Delta\mu_{\text{HB}}$ (Supplementary Fig. 6). Once the oscillations are synchronized, diffusion between lattice sites at equal concentrations is an equilibrium process and does not produce entropy.

In summary, we have introduced the entropy production factor, $\mathcal{E}$, a dimensionless, scalar function that quantifies irreversibility in macroscopic, non-equilibrium dynamics by measuring time-reversal symmetry breaking in the cross-covariances between multiple variables. Integrating $\mathcal{E}$ gives a lower bound on the net entropy production rate, $\dot{s}$. Calculating $\mathcal{E}$ does not require knowledge about the form of the underlying dynamics and is easy to calculate for many types of data, including both random variables, such as the positions of driven colloidal particles[53] (Supplementary Note 9, Supplementary Fig. 8 and 9, Supplementary Table 5), and random fields, such as spatially heterogeneous protein concentrations in cells[54]. Furthermore, we stress that we are only able to resolve the irreversibility present in the observable dynamics of our chemical example. As discussed above, the presence of hidden dynamics will provide underestimates of irreversibility measured via Eq. (1) due to the data processing inequality[55]. Using other observable information, such as asymmetric transition rates[56] or the ratio of populations in observed states under stalled conditions[46] in Markov jump processes, can give tighter bounds on the entropy produced when unobserved, dissipative processes are present. While the examples considered here are simulations of 1+1 dimensional fields, there is nothing inherently different in the methodology if one were to analyze experimental data in 2 or 3 spatial dimensions, such as the 3+1 dimensional time series data attained using lattice-light sheet microscopy[57].

In active matter, both living and non-living, the non-equilibrium dissipation of energy manifests in both time and space. With the method introduced here, compatible with widely-used computational and experimental tools, we provide access to these underexplored modes of irreversibility that drive complex spatiotemporal dynamics.

## Methods

**Calculating $\mathcal{E}$ from data.** Estimate $\mathcal{E}$ requires estimating frequency-space covariance functions, or cross spectral densities (CSDs). Considering a set of $M$ discrete, real variables measured over time: $\{X^i(t)\}$, where $t = \Delta t, \ldots, T$, with $T = N\Delta t$, and $i = 1, \ldots, M$ indexes the variables, we estimate the CSD using the periodogram,

$$\widetilde{C}^{ij}(\omega_n) = \frac{1}{N^2} x^i(\omega_n) x^j(-\omega_n) \tag{12}$$

where $x^i(\omega) = \mathcal{F}\{X^i(t) - \langle X^i(t)\rangle\}$ are the Fourier transforms of the centered variables over the frequencies $\omega_n = 2\pi n T^{-1}$ for $n = [-\frac{N}{2}, \frac{N}{2}]$.

The periodogram, is known to exhibit a systematic bias and considerable variance in estimating the true CSD. Both of these issues can be resolved by smoothing $\widetilde{C}^{ij}$ via convolution with a Gaussian with width $\sigma$. This is equivalent to multiplying $\widetilde{C}^{ij}$ in the time domain by a Gaussian of width $\sigma^{-1}$. We then define our smoothed CSD as

$$\hat{C}^{ij}(\omega_n) = \sum_{\omega_\mu} \Delta\omega \frac{\exp[-(\omega_\mu - \omega_n)^2/2\sigma^2]}{\sqrt{2\pi\sigma^2}} \widetilde{C}^{ij}(\omega_\mu) \tag{13}$$

Once $\hat{C}$ is calculated, we then use the discrete version of Eq. (3) to estimate $\mathcal{E}$. The extension to higher-dimensional data is done as follows: taking into account the spatial lattice on which the data is taken in Eq. (12), convolving the result with a multivariate Gaussian in Eq. (13), and finally estimate $\dot{s}$ using the discrete version of Eq. (4). The choice of smoothing width, $\sigma$, should be guided by the maximum curvature seen in the structure factor, $C^{ij}$[58].

**Bias in $\hat{\mathcal{E}}$ and $\hat{\dot{S}}$.** Our estimates of $\hat{\mathcal{E}}$ and $\hat{\dot{S}}$ are biased. The bias is found by calculating the expected value of $\hat{\dot{S}}$ for a system in equilibrium. To do this, we assume that the true covariance function is $C^{ij} = \delta^{ij}$ and measurement noise plus finite sampling time and rate gives rise to Gaussian noise in both the real and complex parts of $\widetilde{C}^{ij}(\omega)$, obeying the symmetries required for $C^{ij}$ to be Hermitian. We only cite the results here and refer the reader to Supplementary Note 4 for a full derivation. The bias for random variables is

$$\mathcal{E}_{\text{bias}} = \frac{M(M-1)}{2} \frac{\sqrt{\pi}}{T\sigma} \tag{14}$$

$$\dot{S}_{\text{bias}} = \frac{M(M-1)}{2} \frac{\omega^{\text{max}}}{T\sigma\sqrt{\pi}}, \tag{15}$$

where $M$ is the number of variables, $\omega^{\text{max}}$ is the maximum frequency available, $\sigma$ is the width of the Gaussian used to smooth $\widetilde{C}(\omega)$, and $T$ is the total time. The bias for random fields is

$$\mathcal{E}_{\text{bias}} = \left(\frac{M(M-1)}{2} + \frac{3M}{8}\right) \frac{\sqrt{\pi}}{T\sigma_\omega} \prod_{i=1}^{d} \frac{\sqrt{\pi}}{L_i \sigma_{q_i}} \tag{16}$$

$$\dot{s}_{\text{bias}} = \left(\frac{M(M-1)}{2} + \frac{3M}{8}\right) \frac{\omega^{\text{max}}}{T\sigma_\omega\sqrt{\pi}} \prod_{i=1}^{d} \frac{q_i^{\text{max}}}{L_i \sigma_{q_i}\sqrt{\pi}}, \tag{17}$$

where $L_i$ is the length, $q_i^{\text{max}}$ is the maximum wavenumber, and $\sigma_{q_i}$ is the width of the Gaussian used to smooth $\widetilde{C}(\mathbf{q}, \omega)$ in the $i^{th}$ spatial dimension.

**Simulation details.** To simulate the driven Gaussian fields, Eq. (5), we non-dimensionalize the system of equations using a time scale $\tau = (Dr)^{-1}$ and length scale $\lambda = r^{-1/2}$. We use an Euler–Maruyama algorithm to simulate the dynamics of the two fields on a periodic, one-dimensional lattice.

We simulate Eq. (8) using Gillespie's algorithm[59] to create a stochastic trajectory through the $(X, Y)$ phase plane with a well-mixed volume of $V = 100$. We calculate the true $\dot{S}$ of any specific trajectory $\mathbf{z} = \{m_j | j = 1, \ldots, N\}$ as follows. For each state $m'$, there exists a probability per unit time of transitioning to a new state $m$ via a chemical reaction $\mu$, denoted by $W_{m,m'}^{(\mu)}$. At steady state, the true entropy produced is[44]

$$\Delta S_{\text{true}}[\mathbf{z}] = \sum_{j=1}^{N} \ln \frac{W_{m_j, m_{j-1}}^{(\mu_j)}}{W_{m_{j-1}, m_j}^{(\mu_j)}} \tag{18}$$

Note that $\Delta S_{\text{true}}$ is now itself a random variable that depends on the specific trajectory. We estimate $\langle \dot{S}_{\text{true}} \rangle$ by fitting a line to an ensemble average of $\Delta S_{\text{true}}$ (Supplementary Fig. 3), and compare that to $\hat{\dot{S}}$. We calculate $\dot{S}_{\text{blind}}$ by considering the "rate" at which a transition can occur as the sum over all the rates that give rise to the observed transition in $(X, Y)$, i.e.,

$$\Delta S_{\text{blind}} = \sum_{j=1}^{N} \ln \frac{\sum_{\{\mu_j | m_{j-1} \to m_j\}} W_{m_j, m_{j-1}}^{(\mu_j)}}{\sum_{\{\mu_j | m_{j-1} \to m_j\}} W_{m_{j-1}, m_j}^{(\mu_j)}} \tag{19}$$

where $\sum_{\{\mu_j | m_{j-1} \to m_j\}}$ denotes a sum over all reaction pathways $\mu$ that give rise to the transition $m_{j-1} \to m_j$. This procedure coarse-grains $\Delta S_{\text{true}}$, giving $\Delta S_{\text{blind}} \leq \Delta S_{\text{true}}$[47]. $\Delta S_{\text{blind}}$ is the maximum entropy production that can be inferred by any method that observes trajectories in $(X, Y)$, but which does not have access to the reaction pathways followed.

To simulate the reaction-diffusion Brusselator, Eq. (10), we take a compartment-based approach[60] where we treat each chemical species in each compartment as a separate species, and treat diffusion events as additional chemical reaction pathways. We nondimensionalize time by $\tau = (k_1^+)^{-1}$ and use a Gillespie algorithm to simulate all reactions on a one-dimensional periodic lattice with $L$ sites.

See Supplementary Tables 1–5 for all simulation parameters used in each figure.

**Synchronization order parameter**. The synchronization order parameter given in Eq. (11), $r$, is a function of the oscillator phase at every lattice site $j$ at time $t$, $\theta_j(t)$. In order to calculate $\theta$ from our data, we measure the oscillator's phase with respect to a trajectory's mean position over time, namely

$$\theta_j(t) = \arctan\left(\frac{Y_j(t) - \langle Y \rangle}{X_j(t) - \langle X \rangle}\right), \tag{20}$$

where the angle is taken over space and time. This phase is then used to calculate $r$ as given in Eq. (11).

## Data availability

The data that support the findings of this study are available from the corresponding authors upon reasonable request.

## Code availability

The code used to calculate the EPR and EPF from data, as well as run all the simulations in this study, can be found at https://github.com/lab-of-living-matter/freqent.

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

## Acknowledgements

We would like to thank Samuel J. Bryant, Pranav Kantroo, Maria P. Kochugaeva, Rui Ma, Pierre Ronceray, A. Pasha Tabatabai, John D. Treado, Artur Wachtel, Dong Wang, and Vikrant Yadav for insightful discussions. D.S.S. acknowledges support from NSF Fellowship grant #DGE1122492. B.B.M. acknowledges a Simons Investigator award in MMLS and NIH R35 GM138341. M.P.M. and D.S.S. acknowledge support from Yale University Startup Funds. M.P.M. acknowledges funding from ARO MURI W911NF-14-1-0403, NIH RO1 GM126256, Human Frontiers Science Program (HFSP) grant # RGY0073/2018, and NIH U54 CA209992. Any opinion, findings, and conclusions or recommendations expressed in this material are those of the authors and do not necessarily reflect the views of the NSF, NIH, HFSP, or Simons Foundation.

## Author contributions

D.S.S., B.B.M., and M.P.M. conceived the work. D.S.S. wrote and analyzed simulations. All authors contributed to the interpretation of the data and the preparation of the manuscript.

## Competing interests

The authors declare no competing interests.
