## [Peer Review File · Nature Communications]

Reviewers' comments:

Reviewer #1 (Remarks to the Author):

In this theoretical paper, the authors introduce a quantity called entropy production factor (EPF) that is shown to characterize dynamical phase transition and the occurrence of spatial structures in driven systems. Its integral over frequency and wave-vectors reproduces the established overall entropy production rate (EPR). EPF is shown to provide more detailed information than EPR, which, in a sense, is not so surprising since it depends on more variables. The new measure is evaluated for driven Gaussian fields and for the non-linear case of a reaction diffusion system.

The motivating question, which is to find good measures for irreversibility, is a timely and important topic for a broad community ranging from biophysics to statistical mechanics. The new quantity EPF is potentially very useful since it is experimentally accessible. Before I can reach a judgment concerning the significance of the present approach, in particular with regard to earlier work, I would like to see the answer of the authors concerning the following criticism.

1. The status of the integral of EPF as a lower bound to EPR is not clear. While the formulation in abstract and introduction sounds as if this was a hard result, in line 87 it becomes an "approximate lower bound", which I find a strange formulation. If my understanding is correct that this being a bound is just a (certainly interesting) conjecture, it should be disclosed and presented as such.

2. Related: Isn't in the Gaussian case EPF identical to the integrand in the Harada Sasa relation, more precisely, to its field theoretical generalization as introduced in Ref. 28? If this is true, this relation between the two approaches should be shown explicitly.

3. The discussion of the state of the art concerning EPR as a signature of dynamical phase transitions is somewhat misleading, in my opinion.

The authors state (line 32-33) that there is conflicting evidence regarding the continuity of entropy production rates at phase transitions. For well-mixed chemical systems (without explicit spatial dynamics), it has been well established that the first derivative of the entropy production is discontinuous at second-order phase transition (see J. Chem. Phys. 149, 045101 (2018) or Phys. Rev. Lett. 104, 218102 (2010)). At first-order phase transitions (e.g bistability), the rate of entropy production is also known to be discontinuous (see Phys. Rev. Lett. 103, 148103, (2009) and J. R. Soc. Interface 8, 107 (2011)) These nonanalytical behaviors have also been observed for nonequilibrium Ising-like system (see ref. 21 and 23).

So in which sense there is conflicting evidence?

4. Related: The results and discussion regarding the well-stirred Brusselator on page 9 and in the conclusions seem to be in conflict with earlier work J Chem Phys 149, 045101, (2018). There, it has been shown that the first derivative of the entropy production shows a kink at the phase transition. The present authors "do not see clear evidence for that here" as written

in lines 200-203. I do not find this surprising since the authors consider a finite system, for which S can be expected to be continuous. Non-analytical behavior can only occur in the thermodynamic limit. For finite systems, one should rather investigate the finite-size scaling of this derivative as done in the reference just given. Hence, line 247, "the total dissipation rate cannot distinguish between the dynamical phase transitions in the well-mixed and the spatially extended Brusselator" will have to be modified since the statement regarding well-mixed system is in contradiction with the result just cited.

In summary, I recommend a thorough revision that gives a clearer discussion of earlier work, identifies the open questions left from earlier studies, and reaches a more thorough embedding of related approaches. A somewhat fairer description of the merits of EPR at phase transitions will not diminish the potential significance of the more detailed new quantity EPF for spatially extended systems.

Reviewer #2 (Remarks to the Author):

The authors propose a new quantity for characterizing nonequilibrium phase transitions, named entropy production factor (epf), which is closely related to the entropy production rate (epr) (the epr actually corresponds to the integral of the epf over the frequencies. According to them, it would provide more informations about the system.

Stochastic thermodynamics have attracted a great interest in the last years and the behavior of entropy production at phase transitions regimes certainly need further investigations. The paper is interesting and timely and may present an alternative route for such question.

However, it is not clear for me how epr behaves at phase transition regimes. More specifically, what are their signatures at continuous and discontinuous phase transitions? Is there a scaling behavior? How are they compared with the epr? How about its behavior at discontinuous phase transitions?

I am willing for considering the manuscript suitable for publication provided the authors present a description of my such above questions.

Other (minor but important) points :

1. In Figs. 3 and 4, does the phase transition yield at $\Delta \mu_{HB}$?
2. Regarding the comment in 201th-203th lines: The first derivative of epr becoming discontinuous at a continuous phase transitions is not a general finding. Instead, distinct works have show that (actually) its first derivative diverges according to a characteristic critical exponent " α ". In the case when " $\alpha=0$ " (at a mean-field-level), it corresponds to a discontinuity.

Reviewer #3 (Remarks to the Author):

This manuscript studies entropy production and irreversibility of stochastic systems close to a dynamical phase transition. The main result of the work is the introduction of an irreversibility measure [Eqs. (3) and (4)] that provides means to bound the dissipation of a physical system in a nonequilibrium steady state. This measure provides a shortcut to measure dissipation in systems with Gaussian fluctuations, and a lower bound to dissipation when sampling only a subset of all degrees of freedom, or when the noise statistics are non-Gaussian. The authors provide a solid theory supporting these claims, and a plethora of complex numerical simulations for the case of Brusselator models near a Hopf bifurcation. The main novel result resides in the fact that the measure provides further information (namely spatiotemporal) about irreversibility rather than classic measure of Kullback-Leibler divergence rates. I find the paper well-written and interesting, on a timely topic, suitable for a biophysics audience. On the other hand, I have some doubts about its impact to nonequilibrium physics for the reasons I outline below. I have some doubts on whether the article meets the standards of Nature Communications, thus I recommend revision following my comments below in order to clarify its potential impact to a broader audience.

One of my main concerns about the article is the usage of the word dissipation in many instances where the authors are discussing irreversibility, especially because no mention about heat and work is done in the paper. The relation between the KL divergence of mesoscopic degrees of freedom and dissipation has been a topic of intense research in stochastic thermodynamics. Within this field, it has been established that the KL divergence in (1) provides in general a lower bound to dissipation, which is tight when \mathbf{X} does contain all the nonequilibrium degrees of freedom of the system. Otherwise, (1) is just a measure of irreversibility at the level of mesoscopic trajectories. Moreover, the form of (1) does not include reversal of the external protocol, and change of sign of \tilde{X} thus the authors are implicitly assuming a non-equilibrium steady state throughout the dynamics, and the fact that all the observed variables are even under time reversal (e.g. chemical concentrations). This must be said clearly in the paper so readers can know to which systems this result applies and to which do not.

I find the application to systems near a critical point very interesting. However, it must be taken into account that the behaviour of entropy production rates near a critical point have been a subject of previous research. For instance J. Stat. Mech. 11 113207 (2016) discusses the behaviour of entropy production rate close to first and second order phase transitions; arXiv:1803.04743 (2018) discusses entropy production rate close to a Hopf bifurcation, and presents insight on the timescales at which it is observed higher entropy production rate. Similarly Physical Review X 9 (4), 041026 (2019) has introduced spectral decompositions of dissipation in driven liquids. These works are highly related to this manuscript, thus I believe the novelty of the work should be discussed in a more thorough way in order to clarify its suitability for Nature Comms.

In my opinion the idea of measuring irreversibility via an asymmetry property of the Fourier transform of correlation functions could be quite useful in the experimental context. The only limitation that I see to the method is that it requires measurements of $N > 1$ nonequilibrium degrees of freedom, unlike other methods. Thus, it is suitable for systems that may exhibit currents in phase space. For this reason, I believe it would be important to compare the lower bound obtained in Fig 3C and Fig 4C with recently developed bounds based on uncertainty relations of currents [Physical Review Letters, 114(15), 158101 (2015)] and mean first passage times of currents with two absorbing boundaries [Physical Review Letters, 115(25), 250602 (2015)]. There is no knowledge about whether these measures will saturate to s_{blind} or get a tighter bound to the real entropy production, and I believe it would help to grasp better the impact of the measure introduced in this work.

A key concept in the manuscript is the one of the "blind" entropy production. To my understanding this is hard to understand in the main text, and I believe it is mandatory to make it more accessible e.g. by giving examples or by describing what it means with few more lines.

The following sentence in the discussion is not very clear to me "these results suggest that the distribution

of dissipation, not its sum, can uniquely characterise dynamical phase transition in non-equilibrium field theories". What do the authors mean by uniquely?

Reviewer #1 (Remarks to the Author):

In this theoretical paper, the authors introduce a quantity called entropy production factor (EPF) that is shown to characterize dynamical phase transition and the occurrence of spatial structures in driven systems. Its integral over frequency and wave-vectors reproduces the established overall entropy production rate (EPR). EPF is shown to provide more detailed information than EPR, which, in a sense, is not so surprising since it depends on more variables. The new measure is evaluated for driven Gaussian fields and for the non-linear case of a reaction diffusion system.

The motivating question, which is to find good measures for irreversibility, is a timely and important topic for a broad community ranging from biophysics to statistical mechanics. The new quantity EPF is potentially very useful since it is experimentally accessible. Before I can reach a judgment concerning the significance of the present approach, in particular with regard to earlier work, I would like to see the answer of the authors concerning the following criticism.

We thank the reviewer for judging the topic of our paper to be timely and important. We look forward to answering their questions below.

1. The status of the integral of EPF as a lower bound to EPR is not clear. While the formulation in abstract and introduction sounds as if this was a hard result, in line 87 it becomes an "approximate lower bound", which I find a strange formulation. If my understanding is correct that this being a bound is just a (certainly interesting) conjecture, it should be disclosed and presented as such.

At the time of the original submission, the integral of the EPF as a lower bound on the EPR was indeed a conjecture, as we should have been made more clear. Since submission, we have come up with a proof that indeed shows the integral of the EPF to give a lower bound on the EPR. We have clarified the establishment of the EPF's lower bound in the text and added a proof to the supplement, which we add here for the reviewer's convenience:

The KL divergence in Eq. 1 is an exact expression for the entropy production rate provided that the observed set of variables $\{x^\mu\}$ contains every non-equilibrium degree of freedom present in the system. In practice, one only has access to a subset of those degrees of freedom, making the measured KL divergence a lower bound on the entropy production rate. Here, we show that the Gaussian assumption for $P[\mathbf{X}]$ provides another lower bound on the irreversibility measured on the scale of the observed mesoscale trajectories.

The proof relies on the data processing inequality, which states that any transformation of variables $F: x^\mu \rightarrow y^\mu$ will lower the relative entropy between two distributions over both sets variables, i.e. $D_{KL}(P[\{x^\mu\}]||Q[\{x^\mu\}]) \geq D_{KL}(P[\{y^\mu\}]||Q[\{y^\mu\}])$. Intuitively, it states that any processing an observation $\{x^\mu\}$ makes it more difficult to determine whether it came from P or Q . Our strategy will be to choose a transformation that will turn any distribution over x^μ into a Gaussian distribution over y^μ . In our case, our observables are the frequency space variables $x^\mu(\mathbf{q}, \omega)$, and the transformation is a multiplication by a random phase field $\theta(\mathbf{q}, \omega)$, i.e. $x^\mu(\mathbf{q}, \omega) \rightarrow x^\mu(\mathbf{q}, \omega)e^{i\theta(\mathbf{q}, \omega)}$. This random phase, when integrated over frequency space, will make all n -point correlations zero except for the two-point correlation function due to the fact that the variables in real space are real, making the two-point correlation equal to $\langle x^\mu(x^\mu)^* \rangle$,

cancelling the random phase. Thus, the transformed variables are described by a Gaussian distribution (defined as the distribution whose only non-zero cumulants are the first and second), and the data processing inequality guarantees that this provides a lower bound to the KL divergence over the original distributions.

2. Related: Isn't in the Gaussian case EPF identical to the integrand in the Harada Sasa relation, more precisely, to its field theoretical generalization as introduced in Ref. 28? If this is true, this relation between the two approaches should be shown explicitly.

We thank the reviewer for this comment, which prompted us to explicitly show that the EPF for the Gaussian fields is indeed identical to the integrand of the generalized Harada-Sasa relation introduced in the given reference. However, due to the non-conservative dynamics of our Gaussian fields and the non-equilibrium driving coming from a circulating current between the two fields, the EPF is equal to a modified form of the equation given in Nardini *et al.* The modified form is

$$\dot{S} = \int \frac{d\omega}{2\pi} \frac{d\mathbf{q}}{(2\pi)^d} \sigma(\mathbf{q}, \omega); \quad \sigma(\mathbf{q}, \omega) = \frac{\omega}{D} \text{Tr} [\omega C(\mathbf{q}, \omega) - 2\tilde{\mathcal{R}}(\mathbf{q}, \omega)]$$

Where $\tilde{\mathcal{R}}$ is the imaginary part of the response matrix, which has elements $R_{ij} = \delta\langle\eta_i\rangle/\delta h_{\eta_j}$, where $\boldsymbol{\eta} = (\phi, \psi)$, and $C(\mathbf{q}, \omega)$ is the same correlation matrix used to define the EPF. We have added a section to the Supplementary Materials that explicitly shows $\sigma = \mathcal{E}^{\text{DGF}}$ and their equivalence is mentioned in the main text after first defining \mathcal{E}^{DGF} .

3. The discussion of the state of the art concerning EPR as a signature of dynamical phase transitions is somewhat misleading, in my opinion.

The authors state (line 32-33) that there is conflicting evidence regarding the continuity of entropy production rates at phase transitions. For well-mixed chemical systems (without explicit spatial dynamics), it has been well established that the first derivative of the entropy production is discontinuous at second-order phase transition (see J. Chem. Phys. 149, 045101 (2018) or Phys. Rev. Lett. 104, 218102 (2010)). At first-order phase transitions (e.g bistability), the rate of entropy production is also known to be discontinuous (see Phys. Rev. Lett. 103, 148103, (2009) and J. R. Soc. Interface 8, 107 (2011)) These nonanalytical behaviors have also been observed for nonequilibrium Ising-like system (see ref. 21 and 23).

So in which sense there is conflicting evidence?

We thank the reviewer for bringing these works to our attention and they have been cited where appropriate in the revised text. When discussing the conflicting evidence in the field, we had in mind work where spatial dynamics are explicitly considered, namely Falasco *et al.*, Phys. Rev. Lett. 121, 108301 (Ref 22), where the authors do not see any discontinuity of the entropy production rate at the onset of Turing patterns in their (deterministic) equations for the Brusselator. Thus, despite the presence of various non-analyticities in entropy production rates for various systems as the author mentions, there did not seem to be any in a spatially extended systems that exhibit a phase transition. This view is further supported by a preprint that came out during review (Rana and Barato, arXiv:2004.01230), where the authors study the same reaction-diffusion system studied here and state that they “cannot identify any non-analytical behavior of rate of entropy

production σ or its first derivative with respect to $\Delta\mu$ within [their] numerics”, and state that the non-analytic behaviors may instead be present in higher derivatives of the entropy production. We instead show that the non-analytic behavior can be found in \dot{S}_{blind} , which is precisely the quantity estimated by integrating \mathcal{E} .

We have added text to the introducing to clarify our position in the text. We have also further discussed the literature regarding what has been done in the past regarding the behavior of entropy production rates at phase transition points.

4. Related: The results and discussion regarding the well-stirred Brusselator on page 9 and in the conclusions seem to be in conflict with earlier work J Chem Phys 149, 045101, (2018). There, it has been shown that the first derivative of the entropy production shows a kink at the phase transition. The present authors "do not see clear evidence for that here" as written in lines 200-203. I do not find this surprising since the authors consider a finite system, for which S can be expected to be continuous. Non-analytical behavior can only occur in the thermodynamic limit. For finite systems, one should rather investigate the finite-size scaling of this derivative as done in the reference just given. Hence, line 247, "the total dissipation rate cannot distinguish between the dynamical phase transitions in the well-mixed and the spatially extended Brusselator" will have to be modified since the statement regarding well-mixed system is in contradiction with the result just cited.

We again thank the reviewer for bringing our attention to this work, which is indeed relevant for our paper. First, we note that the line mentioned by the reviewer was not referring to the lack of a phase transition in \dot{S} , but rather to the fact that the EPR of the well-mixed and spatially-extended Brusselator are nearly identical, hence the EPR alone cannot distinguish between the 0-dimensional and 1-dimensional systems. We have edited the paragraph in question to clarify our point and added the following figure as Supplementary Figure 6, which plots the true and blinded entropy production rates of the well-mixed and reaction-diffusion Brusselator together, to make it more evident that the two are nearly indistinguishable. The reason for the difference between the two curves for the blinded entropy production rate is also explained in a revised portion of the Discussion section around line 325.

Before discussing our investigation of finite size effects at the recommendation of the reviewer, we note important differences between our work and the one mentioned above. The main difference between our work and that of Nguyen *et al* is the form of the second reaction the definition of the Brusselator (Eq. 1 in their work, Eq. 8 in ours). Namely, our version contains the presence of the intermediate species X and additional chemostat C . This changes the form of the thermodynamic force driving the system away from equilibrium from $\Delta\mu = \ln \frac{k_{-1}k_2[B]}{k_{-2}k_1[A]}$ to $\Delta\mu = \ln \frac{k_{+2}k_{+3}[B]}{k_{-2}k_{-3}[C]}$. In other words, our system is driven away from equilibrium by the chemical potential difference between the chemostats B and C , not between A and B . In addition, one of the lead authors in the paper by Nguyen *et al* recently published a preprint with the same exact dynamics as our Brusselator (Rana and Barato, arXiv: 2004.01230), and they claim that they “cannot identify any non-analytical behavior of rate of entropy production σ or its first derivative with respect to $\Delta\mu$ within [their] numerics”, as mentioned above. The presence of a detectable phase transition in the coarse-grained EPR is itself a result of our work.

We investigate the finite size scaling of the derivative of the entropy production rate. We see that the true entropy production rate does not show any non-analytic behavior as the system size increases, consistent with a recent preprint from one of the authors of the original paper cited by the reviewer (Rana and Barato, arXiv: 2004.01230). Instead we see that the blinded entropy production rate shows a sharper discontinuity as the system volume increases. We found a power-law behavior of $\max(\partial\hat{S}_{\text{blind}}/\partial\Delta\mu) \propto V^{1.22\pm 0.01}$. When we measured the same quantity of our estimated EPR, we still found a power-law behavior, but with a different exponent $\max(\partial\hat{S}/\partial\Delta\mu) \propto V^{1.07\pm 0.05}$. This is due to the fact that, at the value of driving where this maximum occurs nears the bifurcation point, where our Gaussian approximation breaks down most severely. In addition, \hat{S}_{blind} does not exhibit a dependence of V below the transition, and is linearly dependent on V above it.

We further explored why true entropy production does not appear to have any singular behavior while the coarse-grained entropy production does. In the deterministic limit, the EPR is determined by the flux of B molecules turning into C molecules, given by $\dot{S} = \Delta\mu\langle J_{B\rightarrow C} \rangle$, where $\langle J_{B\rightarrow C} \rangle = b\langle x \rangle k_2^+ - c\langle y \rangle k_2^-$. Therefore, in the macroscopic limit, any non-analytic behavior of \dot{S} would arise from the non-analytic behavior of $\langle x \rangle$ or $\langle y \rangle$. $\langle x \rangle = ak_1^+/k_1^-$ is constant, so that leaves $\langle y \rangle$. The Hopf bifurcation of the Brusselator model we study is *supercritical*, which means that the limit cycle grows continuously from the fixed point that becomes unstable at $\Delta\mu_{\text{HB}}$. This can be shown by putting the macroscopic equations of motion in Stuart-Landau Normal form, which is done for our system in *Nat. Comms.* **9**, 1434 (2018). This implies that $\langle y \rangle$ changes continuously as well as the limit cycle emerges. However, the results from the variable transformation only apply when $\Delta\mu - \Delta\mu_{\text{HB}} \ll 1$. Solving the deterministic equations of motion numerically, we see a discontinuous change in $\langle y \rangle$ further from the bifurcation. Nevertheless, this discontinuity is suppressed because driving further from equilibrium results in the forward flux, $J^{\text{F}} = b\langle x \rangle k_2^+$, dominating over the reverse flux, $J^{\text{R}} = c\langle y \rangle k_2^-$. Therefore, the discontinuity in J^{R} becomes difficult to detect in $\dot{S}_{\text{true}} \propto J^{\text{F}} - J^{\text{R}}$ (see panel a). After coarse-graining, the forward flux, $J_{\text{blind}}^{\text{F}} = b\langle x \rangle k_2^+ + \langle x \rangle^3 k_3^-$ and the reverse flux, $J_{\text{blind}}^{\text{R}} = c\langle y \rangle k_2^- + \langle x \rangle^2 \langle y \rangle k_3^+$, become comparable in magnitude, and the discontinuity in the reverse fluxes becomes more apparent.

These results are contained in the new Figure 3 of our manuscript, shown below

We also analyzed the finite size scaling of the EPF. We find that the frequency at which the EPF peaks, ω_{peak} , exhibits a jump in its location that becomes sharper as the volume increases, and the location of the jump approaches the macroscopic bifurcation point, $\Delta\mu_{\text{HB}}$. In addition, the full-width half-maximum of the peak, $\Delta\omega_{\text{FWHM}}$ is maximized near the transition in the location of the peak, reflecting the increased fluctuations of the dynamics near the transition. The size of the width is independent of system size, but the location of the transition changes with system size. The ratio of these first two quantities is an effective quality factor, $Q = \omega_{\text{peak}}/\Delta\omega_{\text{FWHM}}$, whose magnitude is independent of V , with a minimum that moves closer to $\Delta\mu_{\text{HB}}$ as V increases. In addition, the magnitude of the peak of the EPF is independent of V below the transition, and gains a linear dependence on V above it, similarly to \dot{S}_{blind} .

The figure below shows these results and is added as Supplementary Figure 5 in the revised text.

Finally, we turn to the EPF reaction-diffusion Brusselator. In particular, we see that the transition from high- q peaks to low- q peaks for the system gets sharper and occurs closer to the macroscopic transition point as the system size increases. These results are shown in panel b of the figure below, which is in Supplementary Figure 7 in the revised text.

In summary, I recommend a thorough revision that gives a clearer discussion of earlier work, identifies the open questions left from earlier studies, and reaches a more thorough embedding of related approaches. A somewhat fairer description of the merits of EPR at phase transitions will not diminish the potential significance of the more detailed new quantity EPF for spatially extended systems.

Reviewer #2 (Remarks to the Author):

The authors propose a new quantity for characterizing nonequilibrium phase transitions, named entropy production factor (epf), which is closely related to the entropy production rate (epr) (the epr actually corresponds to the integral of the epf over the frequencies). According to them, it would provide more informations about the system.

Stochastic thermodynamics have attracted a great interest in the last years and the behavior of entropy production at phase transitions regimes certainly need further investigations. The paper is interesting and timely and may present an alternative route for such question.

We are glad the reviewer agrees that the topic of the paper is important and relevant. We look forward to addressing their questions below.

However, it is not clear for me how epr behaves at phase transition regimes. More specifically, what are their signatures at continuous and discontinuous phase transitions? Is there a scaling behavior? How are they compared with the epr? How about its behavior at discontinuous phase transitions?

As a point of clarification, we are interpreting the reviewer's first sentence as referring to "how the EPF behaves at phase transition regimes" rather than "how the epr behaves at phase

transition regimes". We refer the reviewer to our responses to Reviewer 1 regarding the scaling behavior of the EPR and EPF, as well as the nature of the transition in our system, which we give numerical evidence for being a discontinuous transition.

I am willing for considering the manuscript suitable for publication provided the authors present a description of my such above questions.

Other (minor but important) points:

1. In Figs. 3 and 4, does the phase transition yield at $\Delta \mu_{\text{HB}}$?

While we do not see any transition in the true entropy production rate, we do see the phase transition in the blinded (or coarse-grained) entropy production rate. The dynamical phase transition, namely the Hopf bifurcation, occurs prior to $\Delta \mu_{\text{HB}}$ due to the finite size of the simulations used in the study. As the system size increases, the transition point, which we take to be the value of $\Delta \mu$ that maximizes $\partial \dot{S} / \partial \Delta \mu$, approaches $\Delta \mu_{\text{HB}}$ (see the figure in the response to point 4 of reviewer 1).

2. Regarding the comment in 201th-203th lines: The first derivative of epr becoming discontinuous at a continuous phase transitions is not a general finding. Instead, distinct works have show that (actually) its first derivative diverges according to a characteristic critical exponent " α ". In the case when " $\alpha=0$ " (at a mean-field-level), it corresponds to a discontinuity.

We thank the reviewer for bringing up this important point. We have edited the text to more fairly discuss what is already known regarding the behavior of \dot{S} at phase transitions.

Reviewer #3 (Remarks to the Author):

This manuscript studies entropy production and irreversibility of stochastic systems close to a dynamical phase transition. The main result of the work is the introduction of an irreversibility measure [Eqs. (3) and (4)] that provides means to bound the dissipation of a physical system in a nonequilibrium steady state. This measure provides a shortcut to measure dissipation in systems with Gaussian fluctuations, and a lower bound to dissipation when sampling only a subset of all degrees of freedom, or when the noise statistics are non-Gaussian. The authors provide a solid theory supporting these claims, and a plethora of complex numerical simulations for the case of Brusselator models near a Hopf bifurcation. The main novel result resides in the fact that the measure provides further information (namely spatiotemporal) about irreversibility rather than classic measure of Kullback-Leibler divergence rates. I find the paper well-written and interesting, on a timely topic, suitable for a biophysics audience. On the other hand, I have some doubts about its impact to nonequilibrium physics for the reasons I outline below. I have some doubts on whether the article meets the standards of Nature Communications, thus I recommend revision following my comments below in order to clarify its potential impact to a broader audience.

We are happy the reviewer finds our work interesting and appreciate the rigor they have clearly applied while reading the paper. We look forward to answering their questions below.

One of my main concerns about the article is the usage of the word dissipation in many instances where the authors are discussing irreversibility, especially because no mention about heat and work is done in the paper. The relation between the KL divergence of mesoscopic degrees of freedom and dissipation has been a topic of intense research in stochastic thermodynamics. Within this field, it has been established that the KL divergence in (1) provides in general a lower bound to dissipation, which is tight when \mathbf{X} does contain all the nonequilibrium degrees of freedom of the system. Otherwise, (1) is just a measure of irreversibility at the level of mesoscopic trajectories. Moreover, the form of (1) does not include reversal of the external protocol, and change of sign of $\tilde{\mathbf{X}}$ thus the authors are implicitly assuming a non-equilibrium steady state throughout the dynamics, and the fact that all the observed variables are even under time reversal (e.g. chemical concentrations). This must be said clearly in the paper so readers can know to which systems this result applies and to which do not.

The reviewer is correct in their comments regarding the important differences between dissipation and irreversibility, especially when hidden degrees of freedom are involved. While our simulations give us access to all the necessary degrees of freedom to tightly bound the dissipation, there is no such guarantee that using the EPF on experimental data will provide that same tightness of the bound. In order to address this, we have changed the text to discuss the relationship between the KL Divergence and the true dissipation, as well as changed the uses of “dissipation” when discussing our simulation results to “irreversibility”.

The reviewer is also correct in our assumption of steady state dynamics of time-reversal symmetric observables. We indeed mention the use of steady state probability distribution functionals in Eq.1 (see line 61), but we appreciate the need to more clearly state all of our assumptions to make them easier to see for readers. We have added a paragraph stating these assumptions in (line 84) the section devoted to deriving the entropy production factor, which we include here for the reviewer’s convenience

It is important to note the assumptions that make our derivation possible. As mentioned above, $P[\mathbf{x}(\omega)]$ describes the dynamics of a non-equilibrium steady state. Further, in writing an expression for $\tilde{P}[\mathbf{x}(\omega)]$, we assumed that the observables are scalar, time-reversal symmetric quantities, such as the chemical concentrations we analyze below.

I find the application to systems near a critical point very interesting. However, it must be taken into account that the behaviour of entropy production rates near a critical point have been a subject of previous research. For instance J. Stat. Mech. 11 113207 (2016) discusses the behaviour of entropy production rate close to first and second order phase transitions; arXiv:1803.04743 (2018) discusses entropy production rate close to a Hopf bifurcation, and presents insight on the timescales at which it is observed higher entropy production rate. Similarly Physical Review X 9 (4), 041026 (2019) has introduced spectral decompositions of dissipation in driven liquids

These works are highly related to this manuscript, thus I believe the novelty of the work should be discussed in a more thorough way in order to clarify its suitability for Nature Comms.

We thank the reviewer for bringing these relevant works to our attention, some of which were known to us and other of which were not. Addressing the differences between our works and those will certainly clarify the novelty of our work.

The work of Zhang and Barato (J. Stat. Mech. 11 113207 (2016)) discusses the behavior of the entropy production rate (and learning rate) of the Ising model subject to an oscillating magnetic field. In the mean field limit, they find that the EPR exhibits a kink at a second-order transition and a discontinuity at a first-order transition, while only evidence for a kink is found in a 2D model. In contrast, we are not claiming to have made a definitive claim on the nature of the EPR at any given transition. Instead, we study a model that does not appear to have any phase transition behavior in the EPR (\dot{S}_{true}), but instead emerges in the coarse-grained EPR (\dot{S}_{blind}), which is what we approximate with our EPR estimate (\hat{S}), and also has signatures in the EPF. In addition, Zhang and Barato characterize true thermodynamic phase transitions, while we are only concerned with a dynamical phase transition.

The work of Roldan *et al* (arXiv:1803.04743 (2018)) discuss a measure of irreversibility from a single degree of freedom by using a whitening transform. While their system (based on oscillations of hair bundles extracted from the ear of a bullfrog) exhibits a Hopf Bifurcation, there is little discussion about the signatures present in \dot{S} as the system approaches the transition, which is a major focus of our work. Further, while our analysis would require a degree of freedom in addition to the location of the tip of the hair bundle, the EPF would provide insight into the frequencies (and therefore timescales) driving the system from equilibrium directly, without resorting to running additional experiments at different sample frequencies, or running the analysis on a various sub-samplings of a single high sampling-rate experiment. In other words, we see our work as complimentary to the work of Roldan *et al*, where one could use their method in the event of access to only a single degree of freedom, while ours gives a more complete picture of the sources of irreversibility in systems of $N \geq 2$ degrees of freedom.

Finally, the work from Tociu *et al* (PRX 9 (4), 041026 (2019)) studies how dissipation is related to diffusion, structure, and biased interactions in a minimal model of active fluids with active tracers embedded in a bath of equilibrium particles. They indeed provide a spectral decomposition for the rate of work done on the active particles, which is derived from the particular form of their model. They also establish a linear relationship between the rate of work done and the integral of the deviation of the non-equilibrium pair correlation function from its equilibrium distribution. This contrasts with the use of the EPF, which is defined for all set of dynamics subject to unknown forces. We further establish our measure of the EPR as a lower bound on the true EPR using the strongly non-linear reaction-diffusion system, establishing the validity of our results far beyond the analytically tractable, minimal models of active colloids in the regime of weak interactions.

In order to better place our work in the context of these and other works brought up by other reviewers, we have added a paragraph in the Discussion section devoted to the works of others and highlighting how ours differs.

In my opinion the idea of measuring irreversibility via an asymmetry property of the fourier transform of correlation functions could be quite useful in the experimental context. The only limitation that I see to the method is that it requires measurements of $N > 1$ nonequilibrium

degrees of freedom, unlike other methods. Thus, it is suitable for systems that may exhibit currents in phase space. For this reason, I believe it would be important to compare the lower bound obtained in Fig 3C and Fig 4C with recently developed bounds based on uncertainty relations of currents [Physical review letters, 114(15), 158101 (2015)] and mean first passage times of currents with two absorbing boundaries [Physical review letters, 115(25), 250602 (2015)]. There is no knowledge about whether these measures will saturate to s_{blind} or get a tighter bound to the real entropy production, and I believe it would help to grasp better the impact of the measure introduced in this work.

Before running these comparisons, it is important to note the regime of their applicability. The bound given by the thermodynamic uncertainty relation depends on an empirical estimate of currents and fluctuations, which is feasible for the well-mixed Brusselator, but is prohibitively difficult for the spatially extended Brusselator due to the curse of dimensionality. The same holds for the mean first passage times of currents with two absorbing boundaries. We therefore can use these alternative measures to compare with the results of Figure 3C, but there is no feasible comparison to be made with the results in Figure 4C, which is one of the benefits of using the spatiotemporal EPF to estimate the EPR.

Nevertheless, we acknowledge the importance of establishing the accuracy of our method with previous ones. We have therefore estimated the EPR using the two methods mentioned by the reviewer. The results are shown below for the well-mixed Brusselator. This figure is added to the text as Supplementary Figure 4 and the explanation below the figure here is in the Supplementary Materials.

The blue, orange, and black dots are the true (\dot{S}_{true}), blind (\dot{S}_{blind}), and estimated (\hat{S}) EPRs found in Figure 3C, where the error bars have been removed from \hat{S} for the sake of clarity. The estimate based on the thermodynamic uncertainty relation (TUR) are shown by the green squares, and the estimate based on mean first-passage times (MFPT) are shown by the red triangles. Both alternative methods also approximate \dot{S}_{blind} because they only take the dynamics in the XY plane into consideration. In addition, they both give lower estimates than \hat{S} , especially in the regime

where the limit cycle emerges. It should be noted that certain choices made when using the alternative methods effects the tightness of the bounds. For those reasons, we outline the specific choices made when using the TUR and MFPT to estimate the entropy production rate below.

To estimating the EPR using the TUR from simulation data, we followed the method outlined in Li *et al*, Nat. Comms. **10**, 1666 (2019). To summarize the method, empirical phase space fluxes $\mathbf{j}(\mathbf{x}, t)$ are integrated over space with a weighting vector field $\mathbf{d}(\mathbf{x})$ to give a “macroscopic current” (j) that accumulates over an observation time,

$$j_{\mathbf{d}}(\tau_{\text{obs}}) = \int_0^{\tau_{\text{obs}}} dt \int d\mathbf{x} \mathbf{j}(\mathbf{x}, t) \cdot \mathbf{d}(\mathbf{x}).$$

The mean and variance of j is then used to estimate the entropy production rate as (setting the Boltzmann constant $k_{\text{B}} = 1$)

$$\dot{S} \geq \dot{S}_{\text{TUR}} = \frac{2\langle j_{\mathbf{d}} \rangle}{\tau_{\text{obs}} \text{Var}(j_{\mathbf{d}})}$$

The choice of weighting field, \mathbf{d} , was shown by Li to significantly affect the tightness of the bound give above. While they devised a Monte-Carlo procedure to minimize the mean-square error of \dot{S}_{TUR} , we opted to use a choice of \mathbf{d} informed by knowledge about the underlying dynamics. Namely, we chose \mathbf{d} to be the macroscopic equations of motion for the Brusselator (i.e. Equation D2 in the supplement).

To estimate the EPR using the MFPT, we follow the reference given by the reviewer, Roldan et al, PRL **115**, 250602 (2015). An observable \mathcal{O} is chosen and the MFPT, $\langle \tau_{\mathcal{O}} \rangle$, for the observable to reach a threshold $L(\alpha) = \ln((1 - \alpha)/\alpha)$ is measured, where α is the fraction of false-positives (and false-negatives) deemed tolerable by the user. The EPR is then bounded by

$$\dot{S} \geq \frac{L(\alpha)(1 - 2\alpha)}{\langle \tau_{\mathcal{O}} \rangle}$$

There exists an optimal choice of \mathcal{O} that minimizes $\langle \tau_{\mathcal{O}} \rangle$ and saturates the above bound, namely the log-likelihood ratio using conditional probabilities of observing a given time series conditioned on the hypothesis that the time series is being played forwards or backwards. In order to only use the dynamics in the XY plane, we chose \mathcal{O} based on the winding number around the center of mass of the observed dynamics, $w(t) \equiv \theta(t)/2\pi$, where $\theta(t) = \arctan\left(\frac{Y-\langle Y \rangle}{X-\langle X \rangle}\right)$, where $\theta(t)$ is measured cumulatively — for example, two full rotations are given by $\theta = 4\pi$. We then make an approximation in treating $w(t)$ as a drift-diffusion process. For a drift-diffusion process, a variable $x(t)$ obeys the Langevin equation $\dot{x} = v + \sqrt{2D}\xi$, where ξ is Gaussian noise. The optimal observable for calculating the MFPT in this case is given by $\mathcal{O}(t) = \frac{v}{D}(x(t) - x(0))$. In this vein, we take our observable to be $\mathcal{O}(t) = \frac{v_w}{D_w}(w(t) - w(0))$, where $v_w = \frac{\langle w(T) \rangle}{T}$ and $D_w = \frac{\text{var}(w(T))}{2T}$, where T is the total observation time and the mean and variance are calculated over many realizations of the same trajectories.

A key concept in the manuscript is the one of the "blind" entropy production. To my understanding this is hard to understand in the main text, and believe it is mandatory to make it more accessible e.g. by giving examples or by describing what it means with few more lines.

We agree with the reviewer that the identification of the “blind” entropy production is indeed an important concept in the paper. We previously defined the blind EPR mathematically in the methods, but we have now added a section to the main text at line 205 to clarify its definition, which we add here for the reviewer’s convenience.

For example, a transition from $(X, Y) \rightarrow (X - 1, Y + 1)$ can occur via reaction k_2^+ or k_3^- in the Brusselator, each of which produces a different amount of entropy in general. Looking only in the (X, Y) plane, it is impossible to tell which reaction took place. When calculating the entropy produced, the rate of making the transition $(X, Y) \rightarrow (X - 1, Y + 1)$ is the sum of the rates for reactions $k_f = k_2^+ + k_3^-$, while the rate of making the reverse transition is the sum of the rates for reactions $k_r = k_2^- + k_3^+$, and the entropy produced after this coarse-graining is $\ln(k_f/k_r)$.

The following sentence in the discussion is not very clear to me "these results suggest that the distribution of dissipation, not its sum, can uniquely characterise dynamical phase transition in non-equilibrium field theories". What do the authors mean by uniquely?

By "uniquely", we meant to highlight the fact that various system configurations can lead to the same EPR but will lead to a different EPF (which is what was meant by the "distribution of dissipation" in the mentioned passage). For example, looking at the EPF and EPR for the driven Gaussian fields (Eq 7), various combinations of α , D , and r can give the same value for the EPR, $\dot{S} = \alpha^2 / (D \sqrt{r})$, but performing a fit of the EPF would allow one to distinguish the form of the underlying potential and driving force. We have edited the paragraph in question to remove this statement in order to make our points clearer.

REVIEWER COMMENTS

Reviewer #1 (Remarks to the Author):

I am satisfied with this thorough revision.

Reviewer #2 (Remarks to the Author):

The authors have made an effort for answering most of referees requests as well as describing relevant remarks under a clearer way. Although the goal of the present manuscript is timely and interesting, some (important) points are not still clear for me. They are listed below :

1. Although the assumption of "gaussian" trajectories is reasonable, authors should state clearer the role parameter ω . In other words, the authors should also exemplify the ω in the two investigated examples.

2. Since the gaussian approach is not exact for the second model, i understand the authors have analyzed other "measures of entropy production" not only as a benchmark of validating such a hypothesis but also for the sake of comparison. I find it is useful, but they should explain/present all of them more clearly and their relationships with the gaussian approach. Also, I do not understand Eq. (18) for evaluating the "blinded" entropy production.

Authors have to describe and compare it more clearly with the true entropy production.

3. In the case of second example (reaction-diffusion Brusselator), the phase transition and their signatures in terms of e_{pr} and e_{pf} have been presented under a very messy way. I strongly suggest authors firstly include an analysis of the model and its phase transition in terms of order-parameter and later to compare them through e_{pr} and e_{pf} signatures. Are fluxes in Fig. 3(c) closely related to the order parameter?

4. As i mentioned in my first report, authors refer to the findings from Ref. [23] as a general finding for the entropy production. Although they constitute important results, they are restricted to the Ising model and continuous phase transitions (such Ising model actually presents a discontinuous phase transitions only in the mean-field-level). On the other hand, Ref. [22] proposed a general description of entropy production trademarks for both continuous and discontinuous phase transitions. In particular, Ref. [22] stated that entropy production does not necessarily diverge or present a singularity in both continuous and discontinuous phase transitions. Conversely, the more general finding concerns the divergence of its first derivative in continuous phase transitions is closely related to the critical exponent α , where for the model considered In Ref. [23] corresponds to a discontinuity (since $\alpha=0$ in the mean-field level). In other words, the apparent general finding is not a discontinuity of its first derivative, but instead an algebraic behavior according to a well defined exponent (see e.g. allied references). Only when $\alpha=0$ in the mean-field level, the the first-derivative of entropy production will present a jump.

I also recomend authors compare the entropy production trademarks (for the second model) with the general findings from Ref. [22], including the finite size scaling analysis in Fig. 5 (supplementary material).

Minor point: 5. In pag 11, authors refer "critical point" to a discontinuous transition. The correct is "transition point" or "coexistence point".

Reviewer #3 (Remarks to the Author):

The authors have successfully replied to all my questions and comments. I recommend publication of the manuscript in Nature Communications in its current form.

REVIEWER COMMENTS

Reviewer #1 (Remarks to the Author):

I am satisfied with this thorough revision.

We thank the reviewer for their positive assessment of our work.

Reviewer #2 (Remarks to the Author):

The authors have made an effort for answering most of referees requests as well as describing relevant remarks under a clearer way. Although the goal of the present manuscript is timely and interesting, some (important) points are not still clear for me. They are listed below:

We thank the reviewer for reiterating the importance of our work. We believe we adequately address their concerns below.

1. Although the assumption of "gaussian" trajectories is reasonable, authors should state clearer the role parameter ω . In other words, the authors should also exemplify the ω in the two investigated examples.

In our work, understanding the role of ω is equivalent to understanding the role of time in a dynamical system. Time is not a parameter that is varied, but rather it parametrizes the trajectories of a dynamical system. As all our correlation functions are measured in Fourier space, ω takes the place of time and parametrizes the trajectories of our dynamical systems in Fourier space. An equivalent relationship exists between the spatial variable \mathbf{x} and the wavevector \mathbf{q} .

We have added text to lines 64 and 107 to clarify these issues.

2. Since the gaussian approach is not exact for the second model, i understand the authors have analyzed other "measures of entropy production" not only as a benchmark of validating such a hypothesis but also for the sake of comparison. I find it is useful, but they should explain/present all of them more clearly and their relationships with the gaussian approach. Also, I do not understand Eq. (18) for evaluating the "blinded" entropy production. Authors have to describe and compare it more clearly with the true entropy production.

We thank the reviewer for valuing our addition of the alternative entropy production rate estimators in response to previous comments from Reviewer 3. We refer Reviewer 2 to Appendix E for a full description of the estimators. In addition, we have added text to the paragraph starting at line 215 to further explain the two estimators, as well as their relationship with our Gaussian approximation. We add the edited text here for the Reviewer's convenience with new text in bold:

To further benchmark our estimator, we calculate \dot{S} using two alternative methods, one based on the thermodynamic uncertainty relation (**TUR**) [7, 47] and one based on measuring first passage times (**MFPT**) [48]. **The prior method measures a macroscopic**

current based on a weighted average of a system's trajectory, j_d , and estimates the EPR using the TUR for diffusive dynamics, $\dot{S} \geq 2 \langle j_d^2 \rangle / \tau_{\text{obs}} \text{Var}[j_d]$, where $\langle \rangle$ and Var denote an ensemble average and variance taken after an observation time τ_{obs} [17]. The latter method requires measuring the MFPT of an observable \mathcal{O} constructed from the system's dynamics to reach a threshold that depends on a user-defined error tolerance. We choose \mathcal{O} and the threshold based on a drift-diffusion approximation for the winding number of the Brusselator (Supplementary Material). Similarly to \hat{S} , both of these methods saturate to the true \dot{S} for systems obeying linear dynamics. As such, they also approximate \dot{S}_{blind} but we find that they provide a looser bound than \hat{S} (Supplementary Figure 4).

Regarding the blind entropy production, we have added text to line 382, just after what was Eq. (18), now Eq. (19), to clarify the notation used in the equation as well as explicitly stating that the \dot{S}_{blind} is a coarse-graining of \dot{S}_{true} and thus $\dot{S}_{\text{true}} \geq \dot{S}_{\text{blind}}$ as per the added citation, Esposito, PRE (2012). We add the edited text here for the Reviewer's convenience, with new text in bold:

We calculate \dot{S}_{blind} by considering the ‘‘rate’’ at which a transition can occur as the sum over all the rates that give rise to the observed transition in (X, Y) , i.e.

$$\Delta S_{\text{blind}} = \sum_{j=1}^N \ln \frac{\sum_{\{\mu_j | m_{j-1} \rightarrow m_j\}} W_{m_j, m_{j-1}}^{(\mu_j)}}{\sum_{\{\mu_j | m_j \rightarrow m_{j-1}\}} W_{m_{j-1}, m_j}^{(\mu_j)}}$$

Where $\sum_{\{\mu_j | m_{j-1} \rightarrow m_j\}} W_{m_j, m_{j-1}}^{(\mu_j)}$ denotes a sum over all reaction pathways μ that give rise to the transition $m_{j-1} \rightarrow m_j$. This procedure coarse-grains ΔS_{true} giving $\Delta S_{\text{blind}} \leq \Delta S_{\text{true}}$ [58]. ΔS_{blind} is the maximum entropy production that can be inferred by any method that observes trajectories in (X, Y) , but which does not have access to the reaction pathways followed.

3. In the case of second example (reaction-diffusion Brusselator), the phase transition and their signatures in terms of epr and epf have been presented under a very messy way. I strongly suggest authors firstly include an analysis of the model and its phase transition in terms of order-parameter and later to compare them through epr and epf signatures. Are fluxes in Fig. 3(c) closely related to the order parameter?

We thank the reviewer for prompting us to characterize our dynamical phase transition using an order parameter before discussing the entropy production. As we make multiple mentions of the transition to *synchronization* separate from the Hopf Bifurcation but had not quantified it, we chose to measure the Kuramoto synchronization order parameter, a standard measure of synchronization in the study of coupled oscillators (for a review, see Acebron *et. al.*, Reviews of Modern Physics (2005)). Specifically, the order parameter r is defined using

$$r e^{i\psi} = \frac{1}{T} \int dt \frac{1}{M} \sum_{j=1}^m e^{i\theta_j(t)}$$

where $\theta_j(t)$ is the phase of the oscillator at position x_j and time t , M is the number of oscillators (here, the number of lattice sites in our simulation), and integral over time is a temporal average.

ψ denotes the overall phase. As explained in a new section of the Methods, we define the phase of the oscillator at each lattice site in terms of the chemical species concentration at that lattice site:

$$\theta_j(t) = \arctan\left(\frac{Y_j(t) - \langle Y \rangle}{X_j(t) - \langle X \rangle}\right)$$

where the average is taken over time and space.

We find r is close to zero in the asynchronous phase ($\Delta\mu < \Delta\mu_{\text{HB}}$) and approaches one as the oscillators synchronize ($\Delta\mu \gtrsim \Delta\mu_{\text{HB}}$). Similarly to the entropy production rate, as the system volume increases, the transition in r approaches a discontinuous transition at $\Delta\mu_{\text{HB}}$. This stems from the discontinuity in Y at $\Delta\mu_{\text{HB}}$ at infinite system size, which is shown by the discontinuous jump in the fluxes shown in Figure 3c (as the Reviewer alluded to). As both \dot{S} and r are functionals of the underlying trajectories, the discontinuity in the trajectories is seen in both \dot{S} and r .

These new results are incorporated in the new Figure 4c, which is shown below.

4. As i mentioned in my first report, authors refer to the findings from Ref. [23] as a general finding for the entropy production. Although they constitute important results, they are restricted to the Ising model and continuous phase transitions (such Ising model actually presents a discontinuous phase transitions only in the mean-field-level). On the other hand, Ref. [22] proposed a general description of entropy production trademarks for both continuous and discontinuous phase transitions. In particular, Ref. [22] stated that entropy production does not necessarily diverge or present a singularity in both continuous and discontinuous phase transitions. Conversely, the more general finding concerns the divergence of its first derivative in continuous phase transitions is closely related to the critical exponent α , where for the model considered In Ref. [23] corresponds to a discontinuity (since $\alpha=0$ in the mean-field level). In other words, the apparent general finding is not a discontinuity of its first derivative, but instead an algebraic behavior according to a well defined exponent (see e.g. allied references). Only when $\alpha=0$ in the mean-field level, the the first-derivative of entropy production will present a jump. I also recomend authors compare the entropy production trademarks (for the second model) with the general findings from Ref. [22], including the finite size scaling analysis in Fig. 5 (supplementary material).

As we show in Figure 3b,c and discuss in the paragraph starting at line 233, the Hopf Bifurcation in our system is *supercritical*, meaning it grows continuously from the previously stable fixed point. However, this continuous growth continues only over a narrow region above the macroscopic bifurcation point, $\Delta\mu_{HB}$. Further from $\Delta\mu_{HB}$, we see a discontinuous transition in the steady state value of the intermediate chemical species Y , which then carries over into the fluxes J which define the entropy production rate by $\dot{S} = J\Delta\mu$ and the synchronization order parameter r , as mentioned above. At the resolution in $\Delta\mu$ used for our stochastic simulations, all our phase transition signatures arise from this discontinuous transition.

Figures 5 and 8 of the cited paper (C. E. Fernández Noa, Pedro E. Harunari, M. J. de Oliveira, and C. E. Fiore, PRE 2019) show a continued sharpening of the transition as the system size gets larger for a system that exhibits a discontinuous transition, similar to what we see in Figure 4c,d in our work. In addition, they see that the EPR as a function of system size crosses a single point for all system sizes, also seen in Figure 1b of Nguyen, Seifert, PRE **102**, 022101 (2020) (Ref 28). We observe the same signature in \dot{S}_{true} . As shown in the image below, \dot{S}_{true} at different system sizes (shown in different colors) cross around $\Delta\mu_{HB}$, most easily seen by the lines switching order to the left and right of the crossing-point.

However, we do not see this behavior in \dot{S}_{blind} . We believe this arises because \dot{S}_{blind} is a coarse-graining of the true EPR, \dot{S}_{true} . To our knowledge, there is no information regarding the preservation of this crossing-point under coarse-graining. Further, we were unable to make a comparison between Ref [22] and Supplementary Figure 5, which shows the scaling of the EPF of the well-mixed Brusselator. There, we find only one feature of the EPF that exhibits any size dependence — the magnitude of the peak, which is independent of system size below the phase transition and gains a volume dependence above it, similarly to the EPR seen in the Figure 3a. As Ref [22] does not study this quantity, we could not find any comparison to be made other than stating that the EPF integrates to the EPR, which agrees with the findings of Ref [22] as discussed above.

In conclusion, we show that we have a discontinuous phase transition with signatures in the EPR that agree with Ref [22]. We have added text to the discussion in line 310 to highlight the general signatures of continuous and discontinuous phase transitions found in Ref [22] and our data's agreement with them.

Minor point: 5. In pag 11, authors refer "critical point" to a discontinuous transition. The correct is "transition point" or "coexistence point".

We have edited the relevant text to “transition point” per the Reviewer’s comment.

Reviewer #3 (Remarks to the Author):

The authors have successfully replied to all my questions and comments. I recommend publication of the manuscript in Nature Communications in its current form.

We thank the reviewer for their positive assessment of our work.

REVIEWERS' COMMENTS

Reviewer #2 (Remarks to the Author):

The authors have answered my questions and remarks properly. I therefore recommend the manuscript for publication.